# Integrable matrix models in discrete space-time

## Žiga Krajnik, Enej Ilievski⋆ and Tomaž Prosen

Faculty of Mathematics and Physics, University of Ljubljana, Slovenia

⋆ enej.ilievski@fmf.uni-lj.si

## Abstract

We introduce a class of integrable dynamical systems of interacting classical matrix-valued fields propagating on a discrete space-time lattice, realized as many-body circuits built from elementary symplectic two-body maps. The models provide an efficient integrable Trotterization of non-relativistic $\sigma$-models with complex Grassmannian manifolds as target spaces, including, as special cases, the higher-rank analogues of the Landau–Lifshitz field theory on complex projective spaces. As an application, we study transport of Noether charges in canonical local equilibrium states. We find a clear signature of superdiffusive behavior in the Kardar–Parisi–Zhang universality class, irrespectively of the chosen underlying global unitary symmetry group and the quotient structure of the compact phase space, providing a strong indication of superuniversal physics.



## Contents



# 1 Introduction

Explaining how macroscopic laws of matter emerge from microscopic reversible dynamics is one the central problems of modern theoretical physics which is still quite far from being settled. One source of difficulties is that dynamical systems which comprise many interacting degrees of freedom are rarely amenable to exact analytic treatment and explicit closed-form solutions are an exception. To make matter worse, numerical simulations of thermodynamic systems out of equilibrium becomes quickly inaccessible at large times owing to exponential growth of required resources, even in one spatial dimension where state-of-the-art methods based on matrix-product states are available. Integrable models provide an opportunity to mitigate some of these issues by providing an ideal theoretical playground and address some key question of statistical physics with a high level of rigour. In spite of a long-lasting progress in the field of classical [1–6] and quantum integrability [7–16], the ultimate hope to obtain explicit solutions to various nonequilibrium problems has not materialized yet, and even the most fundamental question still present a formidable task for analytical methods. Even in the context of classical soliton theories, one of the gems of mathematical physics which culminated with the development of the (inverse) scattering techniques [17–19], neither the direct nor the inverse problem generally permit closed-form solutions, and only rare instances are known where the integration can be carried out in an analytic fashion [20–23]. Indeed, even from a numerical standpoint, no effective framework for computing equilibrium averages of dynamical (or even static) correlation functions is available at this time.

To circumnavigate some of these inherent limitations it is fruitful to attempt a slightly different approach. To better understand certain peculiar features of integrable dynamical systems subject to non-trivial global symmetry constraints, we confine ourselves in this paper to a certain class of classical models in a *discrete* space-time geometry by following the spirit of a preceding work [24]. Our aim is to explore the possibility of realizing simple exactly solvable symplectic circuits which possess conserved non-abelian currents. While sacrificing time-

translational symmetry may at first glance seem an unnecessary hindrance, we wish to argue nonetheless that dynamical systems in discrete time offer certain advantages over Hamiltonian models that can be fruitfully employed in various physics applications. An example of this are simple deterministic cellular automata studied recently in [25, 26, 26–28] which permit one to obtain very explicit results for dynamical correlation functions. In this work, we describe a simple procedure to obtain a class of many-body propagators composed of two-body sympletic maps which governs a discrete space-time evolution of interacting matrix-valued degrees of freedom. This is accomplished in a systematic manner, employing the methods of algebraic geometry and the notion of Lax representation [2] which ensures integrability of the model from the outset. An explicit integration scheme we managed to obtain provides a versatile numerical tool which facilitates efficient numerical simulation of statistical ensembles.

An important source of motivation for this work comes from an ever growing theoretical interest in nonequilibrium phenomena in strongly-correlated quantum systems, nowadays routinely explored in cold-atom experiments using highly-tunable optical lattice setups. Studying systems confined to one spatial dimension is particularly attractive not only because they can exhibit unorthodox phenomena, such as anomalous equilibration [29–34] and anomalous transport laws [35–42], but also thanks to a variety of theoretical tools available to study them. In the past few years, our understanding of transport phenomena in low dimensional systems, both in the linear regime and far from equilibrium, has increased quite dramatically. In the realm of integrable systems, the framework of generalized hydrodynamics [43, 44] has established itself as a versatile analytic and numerical tool which led to universal closed-form expressions for the Drude weights [45–47] and DC conductivities [48–51] and paved the way to many applications [52–65].

This work has been largely inspired by the recent discovery of superdiffusive magnetization transport in the isotropic Heisenberg spin-1/2 chain [66–68], subsequently scrutinized in a number of papers [41, 42, 69], collectively accumulating a convincing numerical evidence for the Kardar–Parisi–Zhang (KPZ) type universality [70] (see also [71, 72] and [73–75]). It is remarkable that the same phenomenon is already visible at the classical level, namely in the integrable classical spin chains symmetric under global $SO(3)$ rotations [24,76,77]. In spite of a phenomenological picture based on an effective noisy Burger's equation proposed recently in [41] and further refined in [42], a complete and quantitative understanding of this curious phenomen is still lacking at the moment. More specifically, aside from partial analytical [37, 39,78] and numerical evidence [79], it is not very clear what is the precise role of non-abelian symmetries and, particularly, if higher-rank symmetries could potentially alter this picture and possibly unveil new types of transport laws. With these questions in mind, we design a class of integrable models whose degrees of freedom are matrix fields which take values on certain compact manifolds. To gain better insight into anomalous nature of spin/charge dynamics in models invariant under the action of non-Abelian Lie groups, we carry out a detailed numerical study of charge transport in maximum entropy states. Our results indicate, quite remarkably, that KPZ scaling is a ubiquitous phenomenon independent of the symmetry structure of the local matrix manifold.

**Outline.**   The paper is structured as follows. In Section 2 we present a novel class of integrable matrix models in discrete space-time. We begin in Section 2.1 by introducing the setting and the zero-curvature formulation, and proceed in Section 2.2 with deriving a two-body symplectic map and the corresponding many-body circuit. Next, in Section 2.3, we detail out various properties of the local phase space and give a concise exposition of complex Grassmannian manifolds along with their symplectic structure (Section 2.3.2). The rest of Section 2 is devoted to various formal apects of our dynamical systems. Since these are not essential for our application, the reader may choose to jump directly to Section 3, where we carry out

a numeric study of charge transport in unbiased maximum-entropy equilibrium and quantify its anomalous character. The interested reader is however warmly invited to read the remainder of Section 2, where we discuss several other key properties of the symplectic map. These include integrability aspects (Section 2.4.1), the space-time self-duality (Section 2.4.2), the Yang–Baxter property (Section 2.4.3), the Hamiltonian representation of the two-body map (Section 2.5) and continuum limits (Section 2.6). In Section 4 we make concluding remarks and outline several open directions. There are five separate appendices which include detailed derivations and additional information on various technical aspects.

# 2 Integrable matrix models

## 2.1 Discrete zero-curvature condition

To set the stage, we shall first introduce the setting. We consider a discrete space-time in the form of a a two-dimensional square lattice. Throughout the paper we adopt the convention that the time flows in the vertical direction and the spatial axis is oriented horizontally towards the right. To each site of the space-time lattice we attach a physical variable. A precise specification of physical degrees of freedom, alongside the associated classical phase space, will be a subject of Section 2.3. By rotating the space-time lattice by $45°$ degrees we introduce the light-cone lattice and assign to each of its vertices (nodes) $(n, m) \in \mathbb{Z}^2$ an *auxiliary* variable $\phi_{n,m}$. *Physical* variables $M_\ell^t$ are situated on the nodes of the space-time lattice $(\ell, t) \in \mathbb{Z}^2$ (at the midpoints of edges of the light-cone lattice) and we label them as $M_{\ell=n+m+1}^{t=n-m}$ (resp. $M_{\ell=n+m+1}^{t=n-m+1}$) when $\ell + t$ is odd (resp. even). We furthermore impose periodic boundary condition in the space direction, that is $M_\ell^t \equiv M_{\ell+L}^t$, assuming the system length $L$ to be even.

The outlined construction rests on the notion of a linear transport problem for the auxiliary variables, see e.g. references [2,5,18,19,80]. Parallel transport along the light-cone directions (i.e. characteristics $\ell \pm t = \text{const}$) reads

$$\phi_{n+1,m} = L_{n,m}^{(+)}(\lambda)\phi_{n,m}, \qquad \phi_{n,m+1} = L_{n,m}^{(-)}(\mu)\phi_{n,m}, \tag{1}$$

where a pair of 'matrix propagators' $L^{(\pm)}$, called the *Lax pair*, represent certain matrix functions of physical variables which additionally depend analytically on the so-called *spectral* parameters $\lambda$ and $\mu$.

The consistency requirement for the above auxiliary linear problem is that the shifts of the light-cone coordinates commute, meaning that moving from $\phi_{n,m}$ to $\phi_{n+1,m+1}$ does not depend on the order of the light-cone propagators. This condition can be neatly encapsulated by the *discrete zero-curvature property* (cf. refs. [81, 82]) around the elementary square plaquette of the light-cone lattice

$$F^{1/2}L^{(+)}(\lambda; M_2)L^{(-)}(\mu; M_1)F^{-1/2} = F^{-1/2}L^{(-)}(\mu; M_2')L^{(+)}(\lambda; M_1')F^{1/2}. \tag{2}$$

Here we have focused on a single plaquette and slightly adapted our notation: the Lax matrices $L^{(\pm)}$ that propagate along the light-cone directions are now functions of local 'edge variables' $M$, whereas $F$ is a constant invertible 'twisting matrix'; primed variables $M'$ are a shorthand notation for $M$ shifted by one unit in the time direction (as depicted in Figure 1). Importantly, the discrete curvature is everywhere satisfied if and only if the updated variables $M_{1,2}'$ are appropriately linked to $M_{1,2}$. Another suggestive interpretation is to think of the flatness condition as a specification of the dynamical propagator, i.e. a local two-body map $(M_1, M_2) \mapsto (M_1', M_2')$ over a spatially adjacent pair of sites $(1, 2)$. In Section 2.4.1 we explain how Eq. (2) gives rise to integrability of the model.

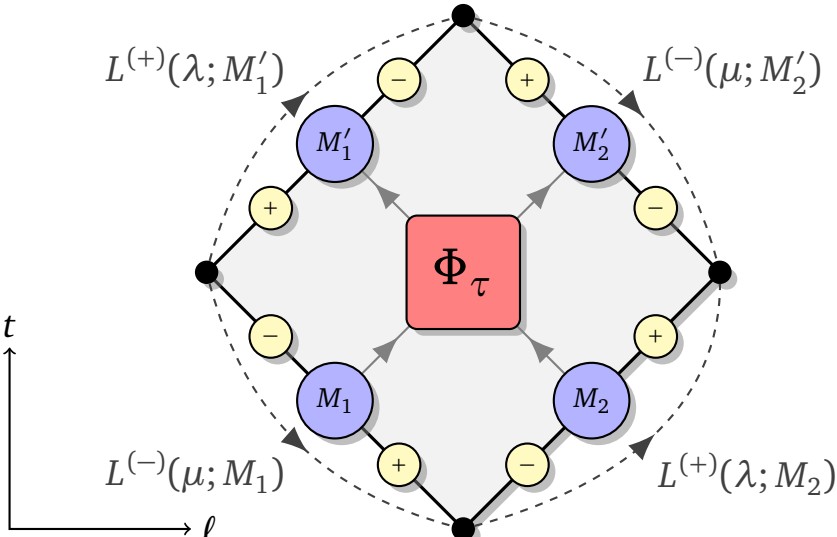

Figure 1: Elementary plaquette of the discrete light-cone lattice: matrix-valued classical fields (blue circles), which belong to a certain manifold, are attached to vertices of the discrete space-time lattice. Primed variables $M'_{1,2}$ pertain to $M_{1,2}$ time-shifted by one unit by application of propagator $\Phi_\tau$. Yellow circles implement local twists of either positive or negative orientation represented by conjugations with constant invertible matrices $F^{1/2}$ and $F^{-1/2}$, respectively.

Obtaining and classifying all physically admissible solutions to Eq. (2) is likely a difficult task and we shall not undertake it in this work. With a more modest goal in mind, we will attempt to find first the simplest solutions by making the following restrictions:

1.  We set both light-cone Lax operators to be equal, $L^{(+)} \equiv L^{(-)}$.

2.  Lax matrix $L(\lambda; M)$ is assumed to be a *linear* function of the spectral parameter $\lambda$.

3.  Lax matrix $L(\lambda; M)$ is assumed to have a *linear* dependence on the matrix variable $M$.

We shall interpret a local physical variable $M$ as a classical matrix field which takes values in $GL(N; \mathbb{C})$ or a submanifold thereof. The third requirement can then be naturally satisfied (without loss of generality) by imposing the *non-linear* constraint

$$M^2 = \mathbb{1}. \tag{3}$$

We make the following ansatz for the Lax matrix complying with (1.-3.),

$$L^{(\pm)}(\lambda; M) \quad \longrightarrow \quad L(\lambda; M) = \lambda \mathbb{1} + \mathrm{i} M, \tag{4}$$

and proceed to look for the solutions of the discrete zero-curvature condition of the form

$$F L(\lambda; M_2) L(\mu; M_1) = L(\mu; M'_2) L(\lambda; M'_1) F. \tag{5}$$

It remarkably turns out that this matrix equation admits a unique non-trivial solution of the difference type, i.e. that there exist a map $(M_1, M_2) \mapsto (M'_1, M'_2)$ depending solely on the difference of the two spectral parameters $\mu - \lambda$. As subsequently demonstrated, the difference condition naturally implies a *dynamical* conservation law

$$M'_1 + M'_2 = F(M_1 + M_2)F^{-1} \equiv \mathrm{Ad}_F(M_1 + M_2), \tag{6}$$

which in the absence of twist ($F = \mathbb{1}$) implies a global conservation law $\sum_\ell M^t_\ell = \text{const}$ when extended to the space-time lattice, see Figure 2.

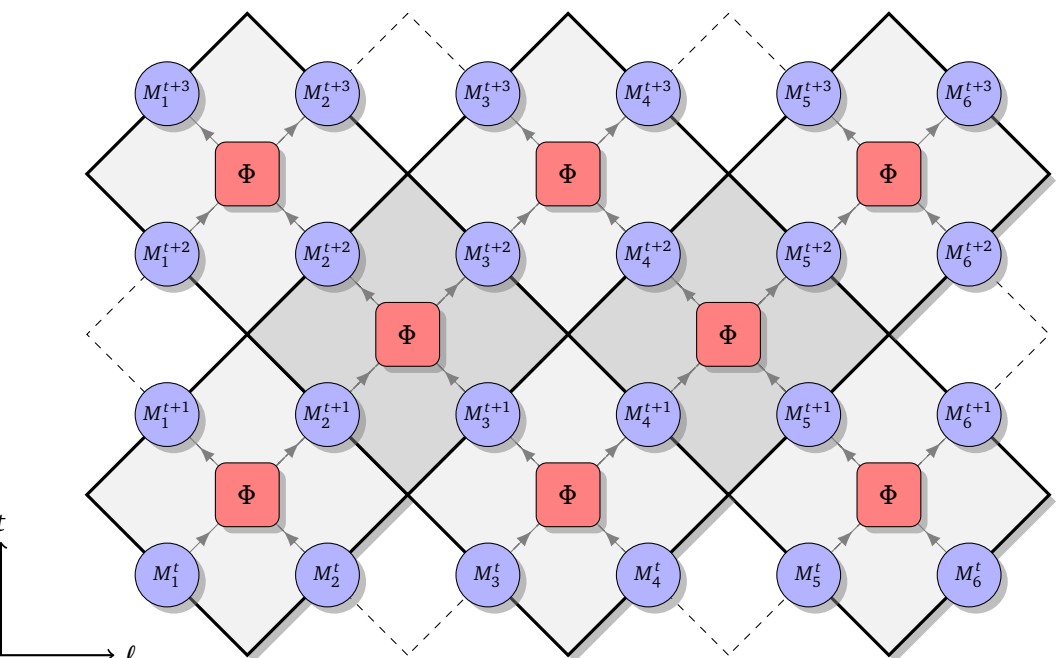

Figure 2: Fabric of discrete space-time: the physical space-time lattice, comprising matrix degrees of freedom $M_\ell^t$ (blue circles), coexisting with the light-cone square lattice depicted by a tilted checkerboard. A two-body symplectic map $\Phi_\tau$ (red square), which is attached to the middle of each tile, provides the time-propagator for every pair of adjacent physical variables.

## 2.2 Dynamical map

The solution to the zero-curvature condition (5), supplemented with nonlinear constraint (3), admits a unique solution of 'difference form' as one-parameter family of *symplectic maps*, $\Phi_\tau : \mathcal{M}_1 \times \mathcal{M}_1 \to \mathcal{M}_1 \times \mathcal{M}_1$,

$$\left(M_1', M_2'\right) = \Phi_\tau(M_1, M_2), \qquad \tau := \mu - \lambda \in \mathbb{R}, \tag{7}$$

representing diffeomorphisms on the product of two manifolds $\mathcal{M}_1$ of involutory matrices. An explicit realization of $\Phi_\tau$ is an adjoint mapping

$$M_1' = \mathrm{Ad}_{FS_\tau}(M_2), \qquad M_2' = \mathrm{Ad}_{FS_\tau}(M_1), \tag{8}$$

generated by an invertible[1] matrix

$$S_\tau \equiv M_1 + M_2 + \mathrm{i}\tau \, \mathbb{1}, \tag{9}$$

where the 'twist field' $F$ can be any constant invertible $GL(N; \mathbb{C})$ matrix. For the proof with a derivation we refer the reader to Appendix A. Note that the map is well defined even for arbitrary complex $\tau$, while we require $\tau$ to be real in this paper in order to allow for its interpretation as a Trotter time step of a Hamiltonian flow.

The above mapping plays a role of the two-site time-propagator (with time-step $\tau$) which provides the basic building block of the many-body symplectic circuit (shown in Figure 2) which, moreover, manifestly preserves the non-linear constraint (3). The above construction is arguably the simplest integrable many-body dynamical system of non-commuting variables in discrete space-time.

---

[1]Non-degeneracy of $S_\tau$ for $|\tau| > 0$ follows from showing $\mathrm{Det}(S_\tau)\mathrm{Det}(S_\tau^\dagger) > 0$, which is a consequence of hermiticity of $M_{1,2}$, commutativity $[M_1 M_2, M_2 M_1] = 0$, and sub-multiplicativity of the operator norm.

**Many-body propagator.** The two-body propagator defined in Eq. (7) constitutes a basic element of the dynamical map $\Phi_\tau^{\text{full}} : \mathcal{M}_L \to \mathcal{M}_L$ defined on the entire phase space $\mathcal{M}_L = \mathcal{M}_1^{\times L}$, denoting the Cartesian product of $L$ copies of $\mathcal{M}_1$.

By virtue of the light-cone structure, the dynamics decomposes into odd and even time-steps,

$$(M_{2\ell-1}^{2t+2}, M_{2\ell}^{2t+2}) = \Phi_\tau(M_{2\ell-1}^{2t+1}, M_{2\ell}^{2t+1}), \qquad (M_{2\ell}^{2t+1}, M_{2\ell+1}^{2t+1}) = \Phi_\tau(M_{2\ell}^{2t}, M_{2\ell+1}^{2t}), \tag{10}$$

respectively, as depicted in Figure 2. With the usual embedding prescription,

$$\Phi_\tau^{(j)} = \underbrace{I \otimes \cdots \otimes I}_{j-1} \otimes \Phi_\tau \otimes \underbrace{I \otimes \cdots \otimes I}_{L-j-1}, \tag{11}$$

where $I : \mathcal{M}_1 \to \mathcal{M}_1$ designates a local unit function $I(M) \equiv M$, the full propagator for a double time step $t \mapsto t+2$ can be split as,

$$\Phi_\tau^{\text{full}} = \Phi_\tau^{\text{even}} \circ \Phi_\tau^{\text{odd}}, \tag{12}$$

with the odd/even propagators further factorizing as

$$\Phi_\tau^{\text{odd}} = \prod_{\ell=1}^{L/2} \Phi_\tau^{(2\ell-1)}, \qquad \Phi_\tau^{\text{even}} = \prod_{\ell=1}^{L/2} \Phi_\tau^{(2\ell)}. \tag{13}$$

In this view, the full map $\Phi_\tau^{\text{full}}$ can be perceived as an 'integrable Trotterization', closely resembling the integrable Trotterization of the quantum Heisenberg model obtained in [83,84]. Indeed, we will shortly demonstrate in Section 2.6 below that Eqs. (8) actually correspond to complete space-time discretizations of (nonrelativistic) $\sigma$-models with compact Lie group cosets as their local target spaces. These include, in particular, the higher-rank analogues of the Landau–Lifshitz magnets [18,85] pertaining to $\mathbb{CP}^n$ manifolds, cf. [86]. This class of classical field theories indeed naturally emerges as the semi-classical limit of *integrable* quantum chains of locally interacting 'spins' which exhibit manifest symmetry under $SU(N)$, introduced a long while ago in [87,88]. In Appendix E we show that the effective classical action governing the long-wavelength modes above a ferromagnetic vacuum yields precisely the continuous space-time counterparts of our matrix models.

## 2.3 Phase space and invariant measures

In this section we proceed by identifying the admissible phase spaces for the dynamical map and describe their formal properties. For a general introduction to differential geometry and symplectic manifolds we refer the reader to one of the standard texts, e.g. [89,90].

In classifying the phase space, the non-linear constraint (3) plays a pivotal role. Moreover, it will be key for us to restrict ourselves to compact smooth manifolds where the notion of a normalizable invariant measure is well defined. This leaves us with the following list of compact Lie groups: (i) unitary groups $U(N)$, (ii) orthogonal groups $O(N)$, and (iii) compact symplectic groups $USp(2N)$.[2]

**Complex Grassmannians.** We begin be examining the unitary Lie groups $G = U(N)$, assuming $N \geq 2$. The first thing to notice is that, by virtue of the involutory property (3), the dynamical matrix variables do not bijectively correspond to the elements of $G$ but instead lie on a submanifold spanned by $N$-dimensional hermitian matrices prescribed by

$$\text{Gr}_{\mathbb{C}}(k, N) := \left\{ M \in GL(N; \mathbb{C}); M^\dagger = M, M^2 = \mathbb{1}, \text{Tr}\, M = N - 2k \right\}. \tag{14}$$

---

[2]There are, in addition, the compact forms of exceptional Lie groups which will be exempted from this study.

This defines the so-called *complex Grassmannian* manifold, the set of $k$-dimensional complex planes embedded in $\mathbb{C}^N$ which pass through the origin, corresponding to the eigenspaces of $M$ with eigenvalue $-1$. Specifically, by prescribing a diagonal *signature matrix*,

$$\Sigma^{(k,N)} = \text{diag}(\underbrace{-1,-1,\ldots,-1}_{k},\underbrace{1,\ldots,1}_{N-k}), \qquad (15)$$

the manifolds $\mathcal{M}_1^{(k,N)} \equiv \text{Gr}_{\mathbb{C}}(k,N)$ can be naturally identified with the adjoint orbits of $\Sigma^{(k,N)}$ under the action of group $G$, that is any $M \in \mathcal{M}_1^{(k,N)}$ can be obtained as

$$M = g \, \Sigma^{(k,N)} g^{\dagger}, \qquad g \in G. \qquad (16)$$

Here we emphasize that a matrix $M \in \mathcal{M}_1$, as given by Eq. (16), does not correspond to a unique group element $g$, the reason being that $\Sigma^{(k,N)}$ is invariant under conjugation with unitary matrices of the form $h \in H = U(k) \times U(N-k)$, that is $\text{Ad}_h \Sigma^{(k,N)} = \Sigma^{(k,N)}$ for all $h \in H$. In this view, $\text{Gr}_{\mathbb{C}}(k,N)$ are *homogeneous spaces*, i.e. cosets of the isometry group $G$ by the stability group $H$,

$$\text{Gr}_{\mathbb{C}}(k,N) \simeq \frac{G}{H} \equiv \frac{U(N)}{U(k) \times U(N-k)} \cong \frac{SU(N)}{S(U(k) \times U(N-k))}, \qquad (17)$$

where $\cong$ stands for diffeomorphic equivalence.[3] The group $G$ acts transitively on each component of $\text{Gr}_{\mathbb{C}}(k,N)$ by virtue of Eq. (16), with the subgroup $H$ being the $G$-stabilizer of $\Sigma^{(k,N)}$. In other words, the group manifold foliates into equivalence classes under the action of $H$, each being an element of the coset space $G/H$. Grassmannian manifolds include, as a special case, complex projective spaces $\text{Gr}_{\mathbb{C}}(1,N) \cong \mathbb{CP}^{N-1}$, representing complex lines passing through the origin of a complex Euclidean space. Due to equivalence $\text{Gr}_{\mathbb{C}}(k,N) \cong \text{Gr}_{\mathbb{C}}(N-k,N)$ we shall subsequently assume, with no loss of generality, that $k \leq \lfloor N/2 \rfloor$.

Matrices which satisfy the involutory property (3) can be alternatively realized in terms of rank-$k$ projectors $P$ projecting onto the eigenspace with eigenvalue $-1$,

$$M = \mathbb{1} - 2P, \qquad P = g \, P_0 \, g^{\dagger}, \qquad P_0 = \begin{pmatrix} \mathbb{1}_k & \mathbb{O}_{k,N-k} \\ \mathbb{O}_{N-k,k} & \mathbb{O}_{N-k} \end{pmatrix}, \qquad (18)$$

where $\mathbb{1}_n$ and $\mathbb{O}_n$ are $n \times n$ unit and zero matrices, respectively, while $\mathbb{O}_{m,n}$ is an $m \times n$ zero matrix.

Complex Grassmannian manifolds $\text{Gr}_{\mathbb{C}}(k,N)$ are preserved (closed) under the map (8), as confirmed by a direct calculation (see Appendix A), so they may be identified with a single-site phase space for our dynamics $\mathcal{M}_1 \simeq \text{Gr}_{\mathbb{C}}(k,N)$.

**Real Grassmannians.** By replacing the unitary Lie group with the (real) orthogonal Lie group $O(N)$, and again demanding $M^2 = \mathbb{1}$, we obtain homogeneous spaces with the coset structure $\text{Gr}_{\mathbb{R}}(k,N) = O(N)/(O(k) \times O(N-k))$, known as real Grassmannian manifolds. A brief inspection shows that real Grassmannians $\text{Gr}_{\mathbb{R}}(k,N)$ are *not* preserved under the dynamical map (8) and hence will not be considered further in this work.

**Lagrangian Grassmannians.** We finally consider complex matrices which preserve the symplectic unit, namely the symplectic group $Sp(2N;\mathbb{C})$. Although the latter is not compact, its restriction to the unitary subgroup, that is the intersection of $Sp(2N;\mathbb{C})$ with $SU(2N)$, is a simply-connected compact Lie group $USp(2N)$ of $2N$-dimensional complex matrices

$$USp(2N) \equiv Sp(N) := \left\{ g \in GL(2N,\mathbb{C}); \quad g^{\dagger} g = \mathbb{1}_{2N}, \quad g^{\mathrm{T}} J g = J \right\}, \qquad (19)$$

---

[3] $S(U(k) \times U(N-k))$ denotes intersection of $SU(N)$ and $U(k) \times U(N-k)$.

called the unitary symplectic group. Here $J$ denotes the standard symplectic unit

$$
J = \begin{pmatrix} \mathbb{0}_N & \mathbb{1}_N \\ -\mathbb{1}_N & \mathbb{0}_N \end{pmatrix} = \mathrm{i}\,\sigma^y \otimes \mathbb{1}_N.
\tag{20}
$$

The adjoint $USp(2N)$ orbits of the *antisymplectic* signature[4]

$$
\Sigma = \mathrm{diag}(\underbrace{1,\ldots,1}_{N},\underbrace{-1,\ldots,-1}_{N}) = \sigma^z \otimes \mathbb{1}_N, \qquad \Sigma^{\mathrm{T}} J\,\Sigma = -J,
\tag{21}
$$

then give antisymplectic unitary involutory matrices $M = g\,\Sigma\,g^{\dagger}$ which satisfy

$$
\mathrm{L}(N) := \left\{ M \in GL(2N,\mathbb{C}); \quad M^2 = MM^{\dagger} = \mathbb{1}, \quad M^{\mathrm{T}} JM = -J \right\}.
\tag{22}
$$

This defines a sub-manifold of $\mathrm{Gr}_{\mathbb{C}}(N, 2N)$ of complex dimension $N(N+1)/2$ known as the complex Lagrangian Grassmannian $\mathrm{L}(N)$, a homogeneous manifold of Lagrangian subspaces in a symplectic vector space of even dimension $2N$ with the quotient structure

$$
\mathrm{L}(N) = \frac{USp(2N)}{U(N)}.
\tag{23}
$$

In Appendix A we demonstrate that the dynamics (8) preserves the Lagrangian submanifold with the anti-symplectic signature, hence Lagrangian Grasmannians again also constitute an admissible phase space $\mathcal{M}_1 \cong \mathrm{L}(N)$.

### 2.3.1 Affine parametrization

To specify a Grassmannian manifold $\mathrm{Gr}_{\mathbb{C}}(k, N)$ one has to supply $k$ linearly independent complex vectors of dimension $N$. Let us suppose these are stored as columns of a complex $N \times k$ matrix $\Psi$. Matrix elements of $\Psi$ are referred to as *homogeneous coordinates* of $\mathrm{Gr}_{\mathbb{C}}(k, N)$. It is crucial to recognize here that the choice of basis vectors is not unique as one enjoys the freedom of performing linear transformations $\Psi \to \Psi A$ with any invertible $k$-dimensional matrix $A$. Given that Grassmannians are identified with equivalence classes $g\,H$, a description in terms of homogeneous coordinates involves (in general non-abelian) gauge freedom. By exploiting this gauge redundancy, we can always pick and choose an element in each equivalence class to bring the coordinate matrix into a 'canonical form'

$$
\Psi = \begin{pmatrix} \mathbb{1}_k \\ Z \end{pmatrix},
\tag{24}
$$

uniquely fixing $(N-k) \times k$ complex matrix coordinates

$$
Z = (z_{i,a}), \qquad i = 1, 2, \ldots, N-k, \quad a = 1, 2, \ldots, k.
\tag{25}
$$

Grassmannian manifolds $\mathrm{Gr}_{\mathbb{C}}(k, N)$ therefore correspond to complex manifolds of *real* dimension $\dim \mathrm{Gr}_{\mathbb{C}}(k, N) = \dim U(N) - \dim(U(k) \times U(N-k)) = 2k(N-k)$ parametrized locally by *affine coordinates* $z_{i,a}$. By parametrizing the group element in the form [91–93]

$$
g(Z) = \begin{pmatrix} (\mathbb{1}_k + Z^{\dagger}Z)^{-1/2} & -(\mathbb{1}_k + Z^{\dagger}Z)^{-1/2}Z^{\dagger} \\ Z(\mathbb{1}_k + Z^{\dagger}Z)^{-1/2} & (\mathbb{1}_{N-k} + ZZ^{\dagger})^{-1/2} \end{pmatrix},
\tag{26}
$$

---

[4]Let us also note that an alternative choice of the signature, $\widetilde{\Sigma} = \mathrm{diag}(1,-1,1,-1\ldots,1,-1) = \mathbb{1}_N \otimes \sigma^z$ would generate symplectic involutory matrices $M$. As shown in Appendix A, this property is not conserved under the dynamics.

projector $P(Z)$ of rank-$k$ assumes the block structure

$$P(Z) = \begin{pmatrix} (\mathbb{1}_k + Z^\dagger Z)^{-1} & (\mathbb{1}_k + Z^\dagger Z)^{-1} Z^\dagger \\ Z(\mathbb{1}_k + Z^\dagger Z)^{-1} & Z(\mathbb{1}_k + Z^\dagger Z)^{-1} Z^\dagger \end{pmatrix}. \tag{27}$$

One shortcoming of such an explicit parametrization is that it does not provide a global parametrization of $\mathcal{M}_1$. Indeed, one needs in total $\binom{N}{k}$ coordinate charts to cover the entire phase space, obtained by all possible distributions of $-1$ in the diagonal signature matrix $\Sigma$.

### 2.3.2 Symplectic structure

Complex Grassmannians $\text{Gr}_{\mathbb{C}}(k, N)$ are symplectic manifolds. They are indeed Kähler manifolds, which means that they possess compatible Riemannian and symplectic structures. In this section we give a succinct review of the basic notions which we subsequently use throughout the rest of the paper.

**Symplectic form.** The local phase space $\mathcal{M}_1^{(k,N)}$ of a matrix model is a smooth manifold endowed with a symplectic 2-from $\omega_K$ which is *closed*, $\mathrm{d}\omega_K = 0$, and *non-degenerate*, $\text{Det}(\omega_K) \neq 0$. Expressed in terms of local affine coordinates $z_{i,a}$ (and their complex conjugates $\bar{z}_{i,a}$) with ranges $i = 1, 2, \ldots, N-k$ and $a = 1, 2, \ldots k$, the Kähler form $\omega_K$ reads compactly

$$\omega_K = \frac{\mathrm{i}}{2} \sum_{i,j=1}^{N-k} \sum_{a,b=1}^{k} \omega_{(i,a),(j,b)} \mathrm{d}z_{i,a} \wedge \mathrm{d}\bar{z}_{j,b}. \tag{28}$$

The Kähler form can be expressed in terms of the *Riemann metric tensor*,

$$\eta(Z) = \left[ \left( \mathbb{1}_{N-k} + ZZ^\dagger \right)^{-1} \otimes \left( \mathbb{1}_k + Z^\dagger Z \right)^{-T} \right], \tag{29}$$

and, employing the vectorized matrix coordinate

$$\mathbf{Z} = \text{vec}(Z) = (z_{1,1}, \ldots, z_{1,k}, z_{2,1}, \ldots, z_{2,k}, \ldots, z_{N-k,1}, \ldots, z_{N-k,k})^{\mathrm{T}}, \tag{30}$$

can be written compactly as $\omega_K = \frac{1}{2\mathrm{i}} \mathrm{d}\mathbf{Z}^\dagger \wedge \eta(Z) \mathrm{d}\mathbf{Z}$. In the special case of complex projective spaces $\mathbb{CP}^n$ one recovers the well-known Fubini–Study metric $\eta_{\text{FS}}$, which can be obtained from the Kähler potential $\mathcal{K} = \log\left(1 + \sum_{j=1}^{n} |z_j|^2\right)$, via $(\eta_{\text{FS}})_{ij} = \partial^2 \mathcal{K} / \partial z_i \partial \bar{z}_j$.

An alternative way of introducing the symplectic structure is to exploit the algebraic structure. This is not only advantageous from the practical standpoint, but also avoids any particular coordinatization of $\mathcal{M}_1^{(k,N)}$. The symplectic form can then be written compactly as

$$\omega = \frac{1}{4\mathrm{i}} \text{Tr}(M \mathrm{d}M \wedge \mathrm{d}M) = -\mathrm{i} \text{Tr}(P \mathrm{d}P \wedge \mathrm{d}P). \tag{31}$$

Here it is crucial that, owing to involutory property $M^2 = \mathbb{1}$, the differentials are subjected to $M \mathrm{d}M + \mathrm{d}M \, M = 0$. It is not hard to explicitly verify that the 2-form given by Eq. (31) is both non-degenerate[5] and closed[6], ensuring that $\omega^{-1}$ exists. Note that the sympectic forms (28) and (31) are equivalent, up to normalization.

---

[5]Since Grassmannians are homogeneous spaces with one connected component, it is sufficient to verify non-degeneracy at one point, e.g. at $\Sigma$.

[6]Closedness can be established immediately: $\mathrm{d}\omega = (4\mathrm{i})^{-1}\text{Tr}(\mathrm{d}M \wedge \mathrm{d}M \wedge \mathrm{d}M) = (4\mathrm{i})^{-1}\text{Tr}(\mathrm{d}M \wedge \mathrm{d}M \wedge \mathrm{d}M \, M^2)$ $= -(4\mathrm{i})^{-1}\text{Tr}(M\mathrm{d}M \wedge \mathrm{d}M \wedge \mathrm{d}M \, M) = -(4\mathrm{i})^{-1}\text{Tr}(\mathrm{d}M \wedge \mathrm{d}M \wedge \mathrm{d}M \, M^2) = -\mathrm{d}\omega = 0$, where we have used $M^2 = \mathbb{1}$ and $M \, \mathrm{d}M + \mathrm{d}M \, M = 0$.

**Vector fields.** Classical observables $f$ are regarded as smooth functions on $\mathcal{M}_1$, $f \in C^\infty(\mathcal{M}_1)$, where for clarity of notation we keep dependence on $k$ and $N$ implicit. The symplectic form $\omega$ provides a mapping from the smooth functions to vector fields via $\mathrm{d}f = \iota_V \omega$, where where $\iota_V \omega$ is the interior product[7]. The vector fields span a complexified tangent plane $\mathcal{T}_M \mathcal{M}_1$ attached to a point $M \in \mathcal{M}_1$. Expanding in the basis of partial derivatives $\partial/\partial z_{i,a}$ and $\partial/\partial \bar{z}_{i,a}$, we can write

$$V = \sum_{i=1}^{N-k} \sum_{a=1}^{k} \left( V_{i,a} \frac{\partial}{\partial z_{i,a}} + \bar{V}_{i,a} \frac{\partial}{\partial \bar{z}_{i,a}} \right). \tag{32}$$

We can nonetheless avoid making any reference to an explicit coordinate system and use the fact that there is a natural action of the group $G = SU(N)$ on $\mathcal{M}_1$ given by conjugation $M \mapsto g M g^\dagger$, where $g \in G$ is a group element of the form $g = \exp\left(-\mathrm{i}\sum_a \theta_a X^a\right)$, with $\theta_a \in \mathbb{R}$ and $X^a \in \mathfrak{g}$ being traceless hermitian matrices generating the Lie algebra $\mathfrak{g} = \mathfrak{su}(N)$. Fixing the basis $\{X^a\}$, $a = 1, 2, \ldots, \dim \mathfrak{g} = N^2 - 1$, with normalization

$$\kappa_{ab} = \mathrm{Tr}(X^a X^b) = \frac{1}{2}\delta_{ab}, \tag{33}$$

the generators satisfy commutation relations[8]

$$[X^a, X^b] = \mathrm{i}\sum_c \epsilon_{abc} X^c, \tag{34}$$

where $\epsilon_{abc}$ is the appropriate tensor of structure constants. Now the matrix-valued vector fields $V_X(M)$ can be viewed as an infinitesimal action of group $G$ at $M \in \mathcal{M}_1$, that is

$$V_X(M) = -\mathrm{i}(\mathrm{ad}X)M \equiv -\mathrm{i}[X, M]. \tag{35}$$

**Momentum maps.** The Lie algebra structure on the space of vector fields realized by the commutator induces a Lie algebra structure on the space of functions provided by the Poisson bracket on the phase space $\mathcal{M}_1$. A mapping from a Lie algebra to functions on classical phase spaces is realized by the *momentum map*, formally obtained by contracting the symplectic form with the vector field

$$\mathrm{d}f_X = \iota_{V_X}\omega. \tag{36}$$

Contracting the symplectic form using $\mathrm{d}M(V_X) = V_X(M)$,

$$\iota_{V_X}\omega = \frac{1}{4\mathrm{i}}\mathrm{Tr}\big(M[V_X(M), \mathrm{d}M]\big), \tag{37}$$

provides the momentum map associated to every generator $X \in \mathfrak{g}$,

$$f_X(M) = \mathrm{Tr}(X M). \tag{38}$$

**Poisson bracket.** The Poisson bracket $\{\cdot, \cdot\}$ is an anti-symmetric bilinear operation which obeys the Liebniz derivation rule and the Jacobi identity, formally defined through the full contraction of the symplectic 2-form.

In any local coordinate chart, the Poisson bracket can be expressed through the inverse of the Riemann metric

$$\{f_1, f_2\}_K = \sum_{j,k} (\eta^{-1})_{jk} \left( \frac{\partial f_1}{\partial z_j} \frac{\partial f_2}{\partial \bar{z}_k} - \frac{\partial f_1}{\partial \bar{z}_j} \frac{\partial f_2}{\partial z_k} \right). \tag{39}$$

---

[7]Contraction of a 2-form $\alpha \wedge \beta$ with a vector field $V$ is computed as $\iota_V(\alpha \wedge \beta) = (\iota_V \alpha)\beta - \alpha(\iota_V \beta)$.

[8]We note that by normalization convention for the generators of $\mathfrak{g}$, we have $\omega = 2\omega_K$.

We again avoid explicit coordinate description by utilizing the momentum maps induced by the action of $\mathfrak{g}$ and accordingly define the Poisson bracket through the contraction of $\omega$,

$$\{f_X, f_Y\} := \omega(V_X, V_Y) = \iota_{V_Y} df_X = \iota_{V_Y} \iota_{V_X} \omega. \tag{40}$$

This readily implies the following relation for the momentum maps,

$$\{f_X, f_Y\} = \frac{1}{4i}\text{Tr}\big(M\big[i[X, M], i[Y, M]\big]\big) = -\text{Tr}\big(i[X, Y]M\big), \tag{41}$$

yielding the Lie–Poisson algebra,

$$\{f_X, f_Y\} = f_{-i[X,Y]} \qquad \Longrightarrow \qquad \{f_{X^a}, f_{X^b}\} = \sum_c \epsilon_{abc} f_{X^c}. \tag{42}$$

Using furthermore that

$$f_{X^a}(M) = \text{Tr}(X^a M) = M^a, \tag{43}$$

we deduce the $\mathfrak{su}(N)$ Lie–Poisson algebra for the hermitian components of $M$,

$$\{M^a, M^b\} = \sum_c \epsilon_{abc} M^c. \tag{44}$$

This is a good place to stress once again that, by virtue of $M^2 = \mathbb{1}$, not all matrix elements (components) $M^a$ can be regarded as independent fields. Indeed, imposing the nonlinear constraint amounts to fixing a symplectic leaf (i.e. Casimir invariants). Consequently, the symplectic form $\omega$ is non-degenerate only on particular adjoint group orbits. With this in mind, the Lie–Poisson bracket (44) on a local phase space $\mathcal{M}_1$ can be more conveniently reformulated in an equivalent matrix form

$$\big\{M \overset{\otimes}{,} M\big\} = -\frac{i}{2}\big[\Pi, M \otimes \mathbb{1}_N - \mathbb{1}_N \otimes M\big], \tag{45}$$

where $\Pi$ is the permutation (transposition, or swap) matrix over $\mathbb{C}^N \otimes \mathbb{C}^N$ and the matrix Poisson bracket is defined as $\big(\{M \overset{\otimes}{,} M\}\big)_{ab,cd} \equiv \{M_{ac}, M_{bd}\}$. This bracket can be immediately lifted to the product phase space $\mathcal{M}_L = \mathcal{M}_1^{\times L}$ by demanding Poisson commutativity at different lattice sites,

$$\big\{M_\ell \overset{\otimes}{,} M_{\ell'}\big\} = -\frac{i}{2}\big[\Pi, M_\ell \otimes \mathbb{1}_N - \mathbb{1}_N \otimes M_{\ell'}\big]\delta_{\ell,\ell'}. \tag{46}$$

Finally, promoting the Lie–Poisson algebra to Lax matrices, the linear bracket (46) can be presented as Sklyanin's fundamental quadratic bracket

$$\big\{L(\lambda; M_\ell) \overset{\otimes}{,} L(\lambda'; M_{\ell'})\big\} = \big[r(\lambda, \lambda'), L(\lambda; M_\ell) \otimes L(\lambda'; M_{\ell'})\big]\delta_{\ell,\ell'}, \qquad r(\lambda, \lambda') = \frac{\Pi}{\lambda' - \lambda}, \tag{47}$$

where the intertwiner $r(\lambda, \lambda')$ is the so-called classical $r$-matrix [12, 18, 19].

**Hamiltonian field.** The Hamiltonian action on a local phase space $\mathcal{M}_1^{(k,N)}$ associated with a vector field $V_\mathcal{H}$ (for Hamiltonian $\mathcal{H} \in \mathfrak{g}$) induces the following dynamics of the momentum maps

$$\frac{d}{dt}f_{X^a}(t) = V_\mathcal{H} f_{X^a} = \{f_{X^a}, f_\mathcal{H}\} = f_{-i[X^a, \mathcal{H}]}. \tag{48}$$

At the level of matrix variables $M$, one can accordingly deduce the 'Heisenberg equation of motion'

$$\frac{d}{dt}M(t) = i[M(t), \mathcal{H}], \tag{49}$$

with the solution

$$M(t) = U_{\mathcal{H}}(t) M(0) U_{\mathcal{H}}^\dagger(t), \qquad U_{\mathcal{H}}(t) \equiv e^{-it\,\mathcal{H}}. \tag{50}$$

Splitting the Hamiltonian $\mathcal{H}$ and the unitary propagator $U_{\mathcal{H}}$ into block form,

$$\mathcal{H} = \begin{pmatrix} \mathcal{A}_k & \mathcal{B}_{k,N-k} \\ \mathcal{B}_{N-k,k}^\dagger & \mathcal{D}_{N-k} \end{pmatrix} \quad \implies \quad U_{\mathcal{H}} = \begin{pmatrix} U_{\mathcal{A}} & U_{\mathcal{B}} \\ U_{\mathcal{C}} & U_{\mathcal{D}} \end{pmatrix}, \tag{51}$$

and using the projector representation of the momentum map,

$$f_{\mathcal{H}}(Z, Z^\dagger) = -2\,\mathrm{Tr}\big(\mathcal{H}\,P(Z, Z^\dagger)\big) = -2\,\mathrm{Tr}\big[(\mathbb{1} + Z^\dagger Z)^{-1}(\mathcal{A} + \mathcal{B}Z + Z^\dagger \mathcal{B}^\dagger + Z^\dagger \mathcal{D}Z)\big], \tag{52}$$

the Hamiltonian equations of motion can also be given in the affine coordinates (cf. Eq. (246) in Section E.1)

$$\frac{dz_j}{dt} = -\mathrm{i}\{z_j, f_{\mathcal{H}}\} = -\mathrm{i}\sum_k \eta_{kj} \frac{\partial f_{\mathcal{H}}}{\partial \bar{z}_k}, \tag{53}$$

along with the complex-conjugate counterpart. These can be recast compactly in the form of a matrix Ricatti equation [91]

$$\frac{d}{dt} Z(t) = \mathrm{i}(Z\mathcal{A} - \mathcal{D}Z + \mathcal{B}^\dagger - Z\mathcal{B}Z). \tag{54}$$

### 2.3.3 Separable invariant measure

Since $\mathcal{M}_1^{(k,N)}$ are homogeneous spaces, they admit a $G$-invariant measure inherited from the invariant Haar measure of the unitary group $G$. This measure is none other than the normalized invariant symplectic volume, defined via the highest exterior product of the Kähler form $\omega_K$ with itself

$$d\Omega^{(k,N)} = \frac{(\omega_K)^{\wedge n}}{n!}, \qquad n = k(N-k). \tag{55}$$

In terms of the Riemann metric tensor we therefore have

$$d\Omega^{(k,N)} = (2\,\mathrm{i})^{-n} \mathrm{Det}(\eta) \prod_{a=1}^{k} \prod_{i=1}^{N-k} d\bar{z}_{i,a} dz_{i,a}. \tag{56}$$

The determinant of the metric tensor can be expressed in terms of the affine matrix coordinate

$$\mathrm{Det}(\eta) = \big[\mathrm{Det}(\mathbb{1}_k + Z^\dagger Z)\big]^{-n}. \tag{57}$$

**Liouville measure.** The Liouville measure specified by density $\rho^{(k,N)}$ refers to an unbiased (flat, or uniform) $G$-invariant probability measure on $\mathcal{M}_1^{(k,N)}$ given by the normalized symplectic volume,

$$\rho^{(k,N)}(M) = \frac{1}{\mathcal{N}^{(k,N)}}, \qquad \mathcal{N}^{(k,N)} = \mathrm{Vol}\big(\mathcal{M}_1^{(k,N)}\big) = \int_{\mathcal{M}_1^{(k,N)}} d\Omega^{(k,N)}. \tag{58}$$

The Liouville volume $\mathcal{N}(k,N)$ can be computed in an indirect manner by exploiting the coset structure of $\mathrm{Gr}_{\mathbb{C}}(k,N)$. The symplectic volume of the unitary group $U(n)$ can easily inferred from the isomorphisms $U(n)/U(n-1) \cong S^{2n-1}$, where the volumes of hypersphere are known to be $\mathrm{Vol}(S^{2n-1}) = 2\pi^n/(n-1)!$, whence

$$\mathrm{Vol}\big(U(n)\big) = \prod_{m=1}^{n} \mathrm{Vol}\big(S^{2m-1}\big) = \prod_{m=1}^{n} \frac{2\pi^m}{(m-1)!} = \frac{2^n \pi^{n(n+1)/2}}{1! \cdots (n-1)!}. \tag{59}$$

Using the coset structure we thus have [94]

$$\mathcal{N}^{(k,N)} = \frac{\mathrm{Vol}(U(N))}{\mathrm{Vol}(U(k))\mathrm{Vol}(U(N-k))} = \frac{(1!\cdots(k-1)!)\pi^{k(N-k)}}{(N-k)!\cdots(N-2)!(N-1)!}. \tag{60}$$

In physics applications, the above measure can be naturally related to the maximum-entropy (or infinite-temperature) equilibrium ensemble of a local matrix degree of freedom. The latter is distinguished by the fact that it maximizes the Shannon/Gibbs entropy

$$\mathfrak{s}[\rho(M)] = -\int_{\mathcal{M}_1^{(k,N)}} \mathrm{d}\Omega^{(k,N)}\rho(M)\log\rho(M). \tag{61}$$

The Liouville measure over the many-body (product) phase space $\mathcal{M}_L^{(k,N)}$ is then given by a product (separable) flat measure

$$\rho_L^{(k,N)}(\{M_\ell\}) = \prod_{\ell=1}^{L} \rho_\ell^{(k,N)}(M_\ell) = \mathrm{const.} \tag{62}$$

To establish that the Liouville measure (62) is invariant under the time evolution generated by $\Phi_\tau^{\mathrm{full}}$, it suffices to verify that the symplectic form $\omega$ (and hence the volume element $\mathrm{d}\Omega$) is preserved under the action of two-body propagator $\Phi_\tau$ given by Eqs. (8). This amounts to show that $\Phi_\tau$ preserves the Poisson bracket, as shown explicitly in Appendix B.

**Grand-canonical measure.** The Liouville measure on $\mathcal{M}_1^{(k,N)}$ admits a multi-parameter extension

$$\rho_\ell^{(k,N)}(M_\ell;\{\mu_b\}) = \frac{1}{\mathcal{Z}^{(k,N)}(\{\mu_b\})}\exp\left[\sum_{a=1}^{\dim\mathfrak{g}} \mu_a f_{X^a}(M_\ell)\right]. \tag{63}$$

The normalization factor

$$\mathcal{Z}^{(k,N)}(\{\mu_b\}) = \int_{\mathcal{M}_1^{(k,N)}} \mathrm{d}\Omega^{(k,N)}\exp\left[\sum_a \mu_a f_{X^a}(M)\right], \tag{64}$$

can be interpreted as the grand-canonical partition function. Formally, this represent a push-forward of the Liouville measure by the momentum map, known in the mathematical literature as an equivariant measure. The class of grand-canonical measures (63) solves the constrained variational problem of entropy maximization (cf. Eq. (61)) with prescribed Lagrange multipliers $\mu_a \in \mathbb{R}$.

Again we can build a product measure over $\mathcal{M}_L$ from the grand-canonical measures over single sites. Since any two adjacent local phase spaces $\mathcal{M}_1 \times \mathcal{M}_1$ in the Cartesian product $\mathcal{M}_L$ are acted on by the group $G$ in a Hamiltonian fashion and the action of $G$ is diagonal, an equivariant measure $\rho_L^{(k,N)}$ on $\mathcal{M}_L$ is also preserved under the action of time-propagator $\Phi_\tau$ provided $F = \mathbb{1}$. This is an immediate corollary of the fundamental conservation law (6).

We can assume, with no loss of generality, that the grand-canonical measure is characterized only by the maximal torus of $G$, namely that the exponent in Eq. (63) is an element $X^a \in \mathfrak{g}_0$ of the maximal Abelian (Cartan) subalgebra $\mathfrak{g}_0$ of $\mathfrak{g}$. The phase-space averages of the Cartan charge densities are given by

$$\langle q^a \rangle = \int_{\mathcal{M}_1^{(k,N)}} \mathrm{d}\Omega^{(k,N)}\rho(M;\{\mu_b\})f_{X^a}(M) = \frac{\partial}{\partial\mu_a}\log\mathcal{Z}^{(k,N)}(\{\mu_b\}), \qquad X^a \in \mathfrak{g}_0. \tag{65}$$

In performing phase-space integrals over an Abelian equivariant measure $\rho^{(k,N)}(M;\{\mu_a\})$ explicit integration can be circumvented thanks to the localization theorem due to Duistermaat and Heckman [95] (see also [96,97]). The statement essentially concerns the exactness of the saddle-point approximation: an integral of the exponent of the Hamiltonian action of a torus group on a compact phase space localizes at its critical points. Recalling the fact that the torus action on $\mathcal{M}_1$ is governed by a linear matrix equation,

$$\frac{\mathrm{d}Z(t)}{\mathrm{d}t} = \mathrm{i}(Z\mathcal{A} - \mathcal{D}Z), \tag{66}$$

where $\mathcal{A}$ and $\mathcal{D}$ are blocks of Hamiltonian matrix (51), with $\mathcal{H} = \sum_a \mu_a X^a$, and assuming for definiteness that the critical points are all isolated (i.e. non-degeneracy $\mathcal{A}_{a,a} \neq \mathcal{D}_{i,i}$ for all $a = 1,\ldots,k$ and $i = 1,\ldots,N-k$), the stationary points $(\mathrm{d}/\mathrm{d}t)Z_\star = 0$ in every coordinate chart are located precisely at the origin $Z_\star = 0$, with the associated Hessian matrix

$$\left.\frac{\partial^2 f_{\mathcal{H}}}{\partial z_{i,a}\partial \bar{z}_{j,b}}\right|_{Z_\star} = \mathcal{A}_{ab}\delta_{ij} - \mathcal{D}_{ij}\delta_{ab}. \tag{67}$$

Denoting $\chi_\alpha = \mathcal{H}_{\alpha\alpha}$ and applying the Duistermaat–Heckman formula one finds (see e.g. [92])

$$\mathcal{Z}^{(k,N)}(\{\chi_\alpha\}) = (-1)^n \pi^{k(N-k)} \sum_\sigma \frac{\exp(\chi_{\sigma_a})}{\prod_{a=1}^k \prod_{\bar{a}\in\bar{\sigma}}(\chi_{\bar{a}} - \chi_{\sigma_a})}, \tag{68}$$

where the summation is over all ordered sets $\boldsymbol{\sigma} = \{\sigma_1 < \sigma_2 < \ldots,\sigma_k\}$ covering all $\binom{N}{k}$ coordinate patches (i.e. all possible redistributions of $-1$'s in the signature $\Sigma^{(k,N)}$), while $\bar{a}$ runs over the complementary set of indices $\bar{\boldsymbol{\sigma}} = \{1, 2, \ldots, N\} \setminus \boldsymbol{\sigma}$.

## 2.4 Integrability structure of the dynamical map

### 2.4.1 Isospectrality

In this section we discuss certain integrability aspects of Eqs. (8) arising from the zero-curvature representation on the discrete two-dimensional light-cone lattice introduced earlier in Section 2.1. Flatness of the Lax connection signifies that parallel transport of the auxiliary variable from one point on the light-cone lattice to another does not depend on the path (Wilson line) between the two; all contractible closed paths on the light-cone lattice are therefore trivial. On the other hand, a discrete holonomy corresponding to a non-contractible path which wraps once around the system (with periodic boundary conditions) is non-trivial and provides an analytic family of matrix-valued functions on the phase space $\mathcal{M}_L$ called the (staggered) *monodromy matrix*

$$\mathbb{M}_\tau(\lambda|\{M_\ell\}) = L(\lambda; M_L)L(\lambda+\tau; M_{L-1})\cdots L(\lambda; M_2)L(\lambda+\tau; M_1), \tag{69}$$

where we have adopted the right-to-left path-ordering convention. The zero-curvature condition implies that any two monodromies with odd or even time argument are related to one another by a similarity transformation. Their eigenvalues, or any spectral invariants, are consequently conserved under the time evolution. Since $\mathbb{M}_\tau(\lambda|\{M_\ell\})$ admits analytic dependence on the spectral parameter $\lambda$, its invariants provide a generating functional for an extensive number (i.e. $\mathcal{O}(L)$) of constants of motion (conservation laws), namely functionally independent phase-space functions preserved under the evolution. This hallmark property of Lax integrability is referred to as *isospectrality* [19,81]. To establish integrability in the Liouville–Arnol'd sense one has to additionally show that these conservation laws are mutually in involution.

The trace of the monodromy matrix defines the *transfer map* $T_\tau(\lambda): \mathcal{M}_L \to \mathbb{C}$,

$$T_\tau(\lambda|\{M_\ell\}) = \mathrm{Tr}\,\mathbb{M}_\tau(\lambda; \{M_\ell\}), \tag{70}$$

constituting a family of mutually Poisson-commuting analytic phase-space functions on $\mathcal{M}_L$

$$\left\{ T_\tau(\lambda; \{M_\ell\}), T_\tau(\lambda'; \{M_\ell\}) \right\} = 0 \qquad \forall \quad \lambda, \lambda' \in \mathbb{C}, \tag{71}$$

where $\tau \in \mathbb{R}$ is a fixed parameter.[9] By sequential application of the local zero curvature relation (5) on even and subsequently odd pairs of Lax operators along a fixed horizontal sawtooth on the light-cone lattice, one obtains time conservation of the transfer map

$$T_\tau(\lambda) \circ \Phi_\tau^{\mathrm{full}} = T_\tau(\lambda). \tag{72}$$

Functions $T_\tau(\lambda)$ then generate, via logarithmic differentiation, a family of *local* conservation laws, at least for signatures with rank $k = 1$. This is shown explicitly in Appendix D.

### 2.4.2 Space-time self-duality

**Dual propagator.** The two-body propagator defined in Eq. (7) realizes the time propagator of the matrix model, namely it propagates a pair of adjacent matrix variables by one unit step along the time direction. The discrete zero-curvature property (5) nonetheless also permits to define the *dual propagator* $\Phi_\tau^{\mathrm{d}}: \mathcal{M}_1 \times \mathcal{M}_1 \to \mathcal{M}_1 \times \mathcal{M}_1$, the spatial analogue of a two-body map where variables $(M_1, M_1')$ are understood as an 'incoming' state and $(M_2, M_2')$ as an 'outgoing' state. See Ref. [98] for a discussion of related concepts in continuum integrable models.

To construct the dual propagator, we consider the linear transport problem for the auxiliary fields in the time direction, noticing that reversing the direction of propagation amounts to inverting the Lax operator (cf. Eq. (4))

$$L(\lambda; M)^{-1} = (\lambda^2 + 1)^{-1} L(\lambda; -M). \tag{73}$$

Starting from Eq. (5), operating by $L(\mu; M_1)^{-1}$ from the right and by $L(\mu, M_2')^{-1}$ from the left, and finally multiplying by twists $F^{-1/2}$ from both sides, we arrive at the *dual* zero-curvature relation (as depicted Figure 3)

$$F^{-1/2} L(\lambda; M_1') F L(\mu; -M_1) F^{-1/2} = F^{-1/2} L(\mu; -M_2') F L(\lambda; M_2) F^{-1/2}. \tag{74}$$

A neat trick to solve this equation is via a local gauge transformation,

$$\widetilde{M}_1 = -M_1, \quad \widetilde{M}_1' = F^{-1} M_1' F, \quad \widetilde{M}_2 = F M_2 F^{-1}, \quad \widetilde{M}_2' = -M_2', \tag{75}$$

which transforms it back to the original form of the (twisted) discrete zero-curvature relation (5)

$$F^{1/2} L(\lambda; \widetilde{M}_1') L(\mu; \widetilde{M}_1) F^{-1/2} = F^{-1/2} L(\mu; \widetilde{M}_2') L(\lambda; \widetilde{M}_2) F^{1/2}, \tag{76}$$

along with the dual dynamical symmetry law (6)

$$(\widetilde{M}_2 + \widetilde{M}_2') = F(\widetilde{M}_1 + \widetilde{M}_1')F^{-1} = \mathrm{Ad}_F(\widetilde{M}_1 + \widetilde{M}_1'). \tag{77}$$

The upshot here is that the local two-body *spatial* propagator in the 'tilde' (gauged) variables exactly coincides with the temporal one,

$$\left( \widetilde{M}_2', \widetilde{M}_2 \right) = \Phi_\tau \left( \widetilde{M}_1', \widetilde{M}_1 \right). \tag{78}$$

---

[9]This statement is a direct corollary of the discrete zero-curvature condition (2) lifted onto the level of transfer maps with the aid of the Sklyanin bracket and Leibniz derivation rule.

An explicit prescription for the dual propagator $\Phi_\tau^{\mathrm{d}}$,

$$(M_2, M_2') = \Phi_\tau^{\mathrm{d}}(M_1, M_1'), \tag{79}$$

can be found by undoing the gauge transformation (75), yielding

$$M_2 = \mathrm{Ad}_{S_\tau^-}(M_1'), \quad M_2' = \mathrm{Ad}_{S_\tau^+}(M_1), \tag{80}$$

where

$$S_\tau^\pm = M_1' F^{\pm 1} - F^{\pm 1} M_1 + \mathrm{i}\tau F^{\pm 1}. \tag{81}$$

In conclusion, the dual (i.e. *spatial*) propagator $\Phi_\tau^{\mathrm{d}}$ and the *temporal* propagator $\Phi_\tau$ (7) are, apart from a local gauge transformation of the plaquette, identical maps.

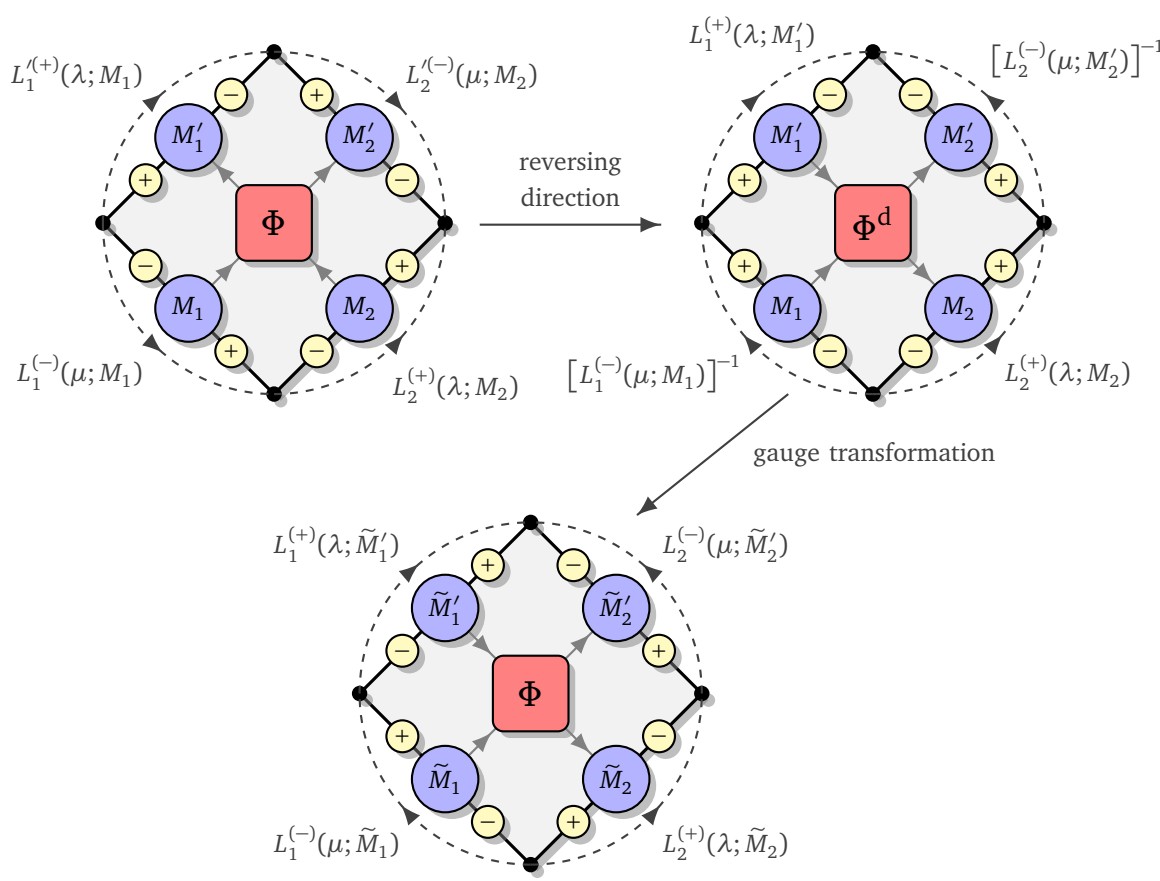

Figure 3: Space-time self-duality: parallel transport in the 'space-like' direction associated with the temporal propagator $\Phi_\tau$ (top left panel) can be interpreted as the 'time-like' parallel transport by inverting the Lax matrices along the 'negative' light-cone axis (top right panel) which defines the dual (i.e. spatial) propagator $\Phi_\tau^{\mathrm{d}}$. The canonical form can be recovered by additionally applying a gauge transformation on the matrix variables (bottom panel), thus establishing a duality between the temporal and spatial propagators.

**Self-duality.** In the absence of twist ($F = \mathbb{1}$), the above gauge transformation can be consistently extended to the entire space-time lattice,

$$\widetilde{M}_\ell^t = (-1)^{\ell + t + 1} M_\ell^t, \tag{82}$$

implying that, in tilde variables, the full *spatial* dynamics can be expressed in terms of the *temporal* propagator

$$(\widetilde{M}_{2\ell}^{2t}, \widetilde{M}_{2\ell}^{2t-1}) = \Phi_{\tau}(\widetilde{M}_{2\ell-1}^{2t}, \widetilde{M}_{2\ell-1}^{2t-1}), \qquad (\widetilde{M}_{2\ell+1}^{2t+1}, \widetilde{M}_{2\ell+1}^{2t}) = \Phi_{\tau}(\widetilde{M}_{2\ell}^{2t+1}, \widetilde{M}_{2\ell}^{2t}). \tag{83}$$

We shall refer to this property as *space-time self-duality*.

Two remarks are in order at this point. First, we wish to point out that the self-duality property, despite its manifest presence in the fully discrete setting, is lost at the level of the Hamiltonian dynamics emerging in the continuous-time limit (cf. Eq. (95)). This is attributed to the fact that space and time coordinates no longer appear on equal footing in the continuous time limit. Indeed, in deriving the continuum limit one only retains smooth variations of the classical field configurations, which is clearly in conflict with the staggered form of the local gauge transformation (75).

It may appear, at the first glance at least, that the self-duality property imposes very stringent restrictions on the dynamics, for instance allowing the temporal and spatial dynamics to be effectively interchanged. This is however *not* the case. We notice that an uncorrelated time-invariant initial state after being locally quenched undergoes a non-trivial time-evolution resulting in a strongly correlated 'time state' [28]. Furthermore, correlations do not only propagate with the unit speed as e.g. in the dual-unitary models [99]. In Section 3 we carry out numerical simulations to explicitly demonstrate this fact.

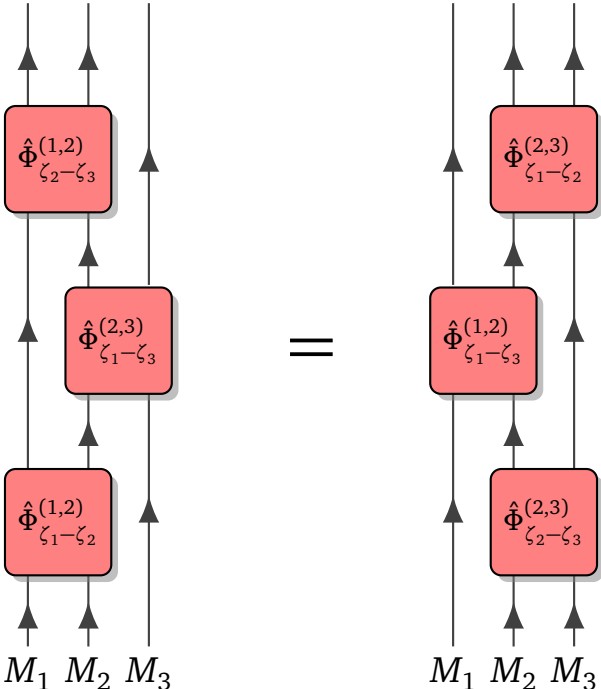

Figure 4: Schematic representation of a set-theoretic (functional) Yang–Baxter equation in the braid form, representing an intertwining property of three consecutive applications of the Yang–Baxter map on a product phase space $\mathcal{M}_1 \times \mathcal{M}_1 \times \mathcal{M}_1$.

### 2.4.3 Yang–Baxter relation

Another important manifestation of integrability (cf. the zero-curvature property (5)) is that the two-body propagator is also a *Yang–Baxter map*, $\mathcal{R}_\lambda : \mathcal{M}_1 \times \mathcal{M}_1 \to \mathcal{M}_1 \times \mathcal{M}_1$, given by

$$\mathcal{R}_\lambda = \Pi \circ \hat{\Phi}_\lambda, \qquad \hat{\Phi}_\lambda = \mathrm{Ad}_{F^{-1} \otimes F^{-1}} \circ \Phi_\lambda = \Phi_\lambda \big|_{F=\mathbb{1}}, \tag{84}$$

where $\hat{\Phi}_\lambda$ denotes the 'untwisted' elementary propagator and $\Pi$ is the permutation map on $\mathcal{M}_1 \times \mathcal{M}_1$. By embedding the maps into a triple Cartesian product $\mathcal{M}_1 \times \mathcal{M}_1 \times \mathcal{M}_1$, we find that $\mathcal{R}_\lambda$ satisfies the *set-theoretic Yang–Baxter relation*

$$\mathcal{R}^{(1,2)}_{\zeta_1-\zeta_2} \circ \mathcal{R}^{(1,3)}_{\zeta_1-\zeta_3} \circ \mathcal{R}^{(2,3)}_{\zeta_2-\zeta_3} = \mathcal{R}^{(2,3)}_{\zeta_2-\zeta_3} \circ \mathcal{R}^{(1,3)}_{\zeta_1-\zeta_3} \circ \mathcal{R}^{(1,2)}_{\zeta_1-\zeta_2}, \tag{85}$$

whereas the untwisted propagator accordingly satisfies the associated braid relation

$$\hat{\Phi}^{(1,2)}_{\zeta_2-\zeta_3} \circ \hat{\Phi}^{(2,3)}_{\zeta_1-\zeta_3} \circ \hat{\Phi}^{(1,2)}_{\zeta_1-\zeta_2} = \hat{\Phi}^{(2,3)}_{\zeta_1-\zeta_2} \circ \hat{\Phi}^{(1,2)}_{\zeta_1-\zeta_3} \circ \hat{\Phi}^{(2,3)}_{\zeta_2-\zeta_3}. \tag{86}$$

Let us briefly elucidate the origin of the Yang–Baxter map (see [81, 100–102], or [103, 104] for more recent accounts which discuss its connection to quasitriangular Hopf algebras [105, 106]). To this end it is convenient to regard the discrete zero-curvature condition (5) as a re-factorization problem [107] for a pair of Lax matrices

$$L(\lambda + \zeta_2; M_2')L(\lambda + \zeta_1; M_1') = L(\lambda + \zeta_1; M_1)L(\lambda + \zeta_2; M_2), \tag{87}$$

where $\zeta_j \in \mathbb{C}$ are arbitrary shift parameters. The Yang–Baxter map $\mathcal{R}_\lambda$ provides a mapping $(M_1, M_2) \mapsto (M_1', M_2')$ which is a *unique* solution to the matrix re-factorization problem. The set-theoretic (functional) Yang–Baxter property is a statement about equivalence of two different intertwining protocols; applying the left- and right-hand sides of Eq. (85) to the sequence $L(\lambda_1; M_1)L(\lambda_2; M_2)L(\lambda_3; M_3)$, where $\lambda_\ell \equiv \lambda + \zeta_\ell$, we obtain

$$L(\lambda_3; M_3^\circ)L(\lambda_2; M_2^\circ)L(\lambda_1; M_1^\circ) \quad \text{and} \quad L(\lambda_3; M_3^\bullet)L(\lambda_2; M_2^\bullet)L(\lambda_1; M_1^\bullet), \tag{88}$$

respectively. Here $M_\ell^\circ$ and $M_\ell^\bullet$, with $\ell = 1, 2, 3$, are two (apriori distinct) sets of propagated variables. Firstly, by uniqueness of matrix re-factorization (87), each application of the Yang–Baxter map preserves the cubic $\lambda$-polynomial, and hence the two expressions in (88) must be equal. To establish the Yang–Baxter property (85) it is left to prove that factorization of a given $\lambda$-polynomial into an ordered product of Lax matrices is unique. This is to say that all the variables are pairwise equal, $M_\ell^\circ = M_\ell^\bullet$ for all $\ell$. Although we suspect that this assertion can be resolved on a formal basis[10], at this moment we are only able to give an explicit algebraic proof (see Appendix A.1).

## 2.5 Symplectic generator

Having shown that Eq. (7) provides a symplectic transformation, we can alternatively realize it in the Hamiltonian form. To this end we define

$$\frac{\mathrm{d}}{\mathrm{d}t}M_\ell = \{M_\ell, \mathscr{H}^{(k,N)}_\tau\}, \tag{89}$$

and require that for both $\ell = 1, 2$ at time $t = \tau$ the solution to Eq. (89) yields the symplectic map (7), namely $M_{1,2}(t = \tau) = M_{1,2}'$. Beware that $\mathscr{H}^{(k,N)}_\tau$ is not simply an integrable lattice Hamiltonian obtained in the $\tau \to 0$ limit (derived below in Section 2.6). It is also important

---

[10]To begin with, uniqueness of factorization for general system size is crucial for well-posedness of the inverse scattering transform.

to stress that the notion of energy is not meaningful here due to broken time-translational symmetry. In this respect, the sought-for generating function $\mathscr{H}_\tau^{(k,N)}$ should be understood merely as an auxiliary quantity (which need not be necessarily a real function, in general). The only physical requirement, besides generating the symplectic map (8), is that in the limit of continuous time the generator $\mathscr{H}$ indeed yields a real (integrable) Hamiltonian function.

For definiteness we confine ourselves here to the untwisted case and set $F = \mathbb{1}$. The generator $\mathscr{H}$ is in general a certain functional of trace invariants of the two-body matrix $S_0 \equiv M_1 + M_2$, that is scalars $s_m = \mathrm{Tr}(S_0^m)$. Taking into account that $s_0 = N$, $s_1 = 2(N - 2k)$, and that all odd invariants $s_{2m+1}$ are proportional to $s_1$ as a consequence of cyclicity of the trace and $M_{1,2}^2 = \mathbb{1}$, it is thus sufficient to retain only $s_{2m}$ for $m \in \mathbb{N}$. We have succeeded in deriving a system of PDEs that determines the generators $\mathscr{H}_\tau^{(k,N)}(\{s_{2m}\})$, and below we give a short summary of the main results. The full derivation is relegated to Appendix B.

We have not managed to obtain a compact and completely general solution to these equations. Using the knowledge of the explicit solutions for matrices of dimension two and four, we instead put forward a *conjecture* for the general form in the simplest case of even dimensional traceless matrices ($N \in 2\mathbb{N}$, $k = N/2$)

$$\mathscr{H}_\tau^{(N/2,N)}(\{s_{2m}\}) = \sum_{j=1}^{N/2} \left[ \log\left(\tau^2 + \tilde{s}_j^2\right) + \frac{2\tilde{s}_j}{\tau} \arctan\left(\frac{\tau}{\tilde{s}_j}\right) \right], \tag{90}$$

parametrized by the *double roots* $\tilde{s}_j$ of the associated Cayley–Hamilton polynomial $\mathrm{p}_{k,N}(\xi) = \mathrm{Det}(\xi\mathbb{1}_N - S_0)|$ (for $k = N/2$, i.e. $s_1 = 0$), that is $\mathrm{p}_{N/2,N}(\tilde{s}_j) = \mathrm{p}'_{N/2,N}(\tilde{s}_j) = 0$. The generators associated to matrices of odd dimension take a slightly different form and are generally complex-valued. In the $N = 3$ case for instance, the generator

$$\mathscr{H}_\tau^{(1,3)} = \frac{2}{3}\left[ \log\left(\tilde{s}^2 + \tau^2\right) + \frac{2\tilde{s}}{\tau} \arctan\left(\frac{\tau}{\tilde{s}}\right) \right] + \mathrm{i}\left(\frac{\tilde{s}^2 + 2}{6\tau}\right)\log\left(1 - \mathrm{i}\tau/2\right), \tag{91}$$

with $\tilde{s}$ defined through $\mathrm{p}_{1,3}(\tilde{s}) = 0$, acquires an extra purely imaginary part. The latter nonetheless vanishes in the limit $\tau \to 0$, which we expect to be a general feature of symplectic generators in odd-dimensional cases.

With aid of the Vieta's formulas and the Jacobi identity we can moreover infer the local Hamiltonian in the continuous time limit, again for even $N$:

$$\lim_{\tau \to 0} \mathscr{H}_\tau^{(N/2,N\in 2\mathbb{Z})} = \log \prod_{j=1}^{N} \tilde{s}_j^2 = \log \mathrm{Det}(M_1 + M_2) = \mathrm{Tr}\log(M_1 + M_2). \tag{92}$$

## 2.6 Semi-discrete and continuum limits

To elucidate the physical meaning of our matrix models it it is instructive to also inspect their time-continuous and field-theoretical limits. We consider first the limit $\tau \to 0$, where the symplectic map $\Phi_\tau$ 'smoothens out' into an integrable Hamiltonian flow governed by a lattice Hamiltonian $H_{\text{lattice}}$. For this purpose we parametrize the twist field as $F = \exp(-\mathrm{i}\tau B/2)$, with $B \in \mathfrak{g}$, and expand Eq. (7) to the lowest order in $\tau$. This yields the differential-difference equation

$$\frac{\mathrm{d}M_\ell}{\mathrm{d}t} = \left\{M_\ell, H_{\text{lattice}}\right\} = -\mathrm{i}\left[M_\ell, (M_{\ell-1} + M_\ell)^{-1} + (M_\ell + M_{\ell+1})^{-1} + B\right], \tag{93}$$

which is generated by the following lattice Hamiltonian[11]

$$H_{\text{lattice}} = \sum_{\ell=1}^{L} \left( \text{Tr}(M_\ell B) - \text{Re Tr} \log(M_\ell + M_{\ell+1}) \right). \tag{94}$$

Notice that $\text{Tr} \log(M_1 + M_2)$ indeed matches $\log \text{Det}(M_1 + M_2)$ obtained in the previous section, see Eq. (92). The obtained equation of motion can be perceived as an integrable non-relativistic sigma model (with an applied external field $B$) on a lattice with variables taking values on cosets $G/H$. The special (rank $k = 1$) case of complex projective planes $\mathbb{CP}^{N-1}$ represent generalized (higher-rank) lattice Landau–Lifshitz models.

Finally, we inspect the field-theory limit of the matrix models by retaining only smooth configurations in the spatial direction. To this end we reintroduce the lattice spacing $\Delta$ and expand a smoothly varying matrix field $M_\ell(t) \to M(x = \ell\Delta, t)$ as $M_{\ell+\Delta} \to M + \Delta M_x + (\Delta^2/2)M_{xx} + \mathcal{O}(\Delta^3)$. For notational convenience we shall write $f_t \equiv \partial_t f$, $f_x = \partial_x f$, and similarly for derivatives of higher order. Sending $\Delta \to 0$ whilst simultaneously rescaling time $t \to (2/\Delta^2)t$ and the magnetic field strength $B \to (\Delta^2/2)B$, we arrive at a family of integrable PDEs of the form

$$M_t = \{M(x, t), H_c\} = \frac{1}{2\mathrm{i}} \left[ M, M_{xx} \right] + \mathrm{i}[B, M]. \tag{95}$$

The latter is generated by the continuum counterpart of Eq. (94),

$$H_c = \int \mathrm{d}x \left[ \frac{1}{4} \text{Tr}(M_x^2) + \text{Tr}(M B) \right]. \tag{96}$$

The Poisson bracket for this field theory is found by taking the continuum limit of the linear Poisson bracket (46),

$$\left\{ M(x) \overset{\otimes}{,} M(x') \right\} = -\frac{\mathrm{i}}{2} \left[ \Pi, M(x) \otimes \mathbb{1}_N - \mathbb{1}_N \otimes M(x) \right] \delta(x - x'). \tag{97}$$

**Lax equations.** The auxiliary linear transport problem for the auxiliary field $\phi_\ell(t)$ in discrete space and continuous time takes the form

$$\partial_t \phi_\ell(t) = V_\ell(\lambda)\phi_\ell(t), \qquad \phi_{\ell+1}(t) = L_\ell(\lambda)\phi_\ell(t), \tag{98}$$

where the spatial propagator $L_\ell$ is the Lax matrix (4) inherited from the light-cone lattice and the temporal component $V_\ell$ we now determine below. The compatibility condition for Eqs. (98) takes the form of a *semi-discrete* zero-curvature condition,

$$\frac{\mathrm{d}}{\mathrm{d}t} L_\ell(t) = V_{\ell+1}(t)L_\ell(t) - L_\ell(t)V_\ell(t), \tag{99}$$

which can be deduced by taking the time derivative of the second equation in Eqs. (98) and combining it with the first equation. It is clear from Eq. (93) that the temporal component of the connection $V_\ell$ acts non-identically on a pair of adjacent lattice sites $\ell$ and $\ell - 1$, i.e. it depends on variables $M_{\ell-1}$ and $M_\ell$. An explicit form can be inferred from the equation of motion $(\mathrm{d}/\mathrm{d}t)\mathrm{L}_\ell(t) = \{L_\ell(t), H_{\text{lattice}}\}$, which, after some algebraic exercising (cf. Appendix C) yields

$$V_\ell(\lambda) = \frac{-2\lambda}{1 + \lambda^2} (L_\ell(0) + L_{\ell-1}(0))^{-1} L_\ell(\lambda^{-1}) + \mathrm{i}B. \tag{100}$$

---

[11]The equation of motion can be inferred directly with help of the linear Poisson bracket (46), yielding the equation of motion given by Eq. (95), for any pair $(k, N)$.

We finally obtain the Lax connection for the continuum counterpart. Reintroducing the lattice spacing parameter $\Delta$ and expanding Eq. (99) to the second order $\mathcal{O}(\Delta^2)$, we obtain the auxiliary linear transport problem associated to a differentiable manifold,

$$\partial_x \phi(x,t) = \mathscr{U}(\lambda; x, t)\phi(x, t), \qquad \partial_t \phi(x, t) = \mathscr{V}(\lambda; x, t)\phi(x, t), \tag{101}$$

satisfying the zero-curvature compatibility condition

$$\partial_t \mathscr{U} - \partial_x \mathscr{V} + [\mathscr{U}, \mathscr{V}] = 0, \tag{102}$$

with connection components

$$\mathscr{U}(\lambda; x, t) = \frac{\mathrm{i}}{\lambda} M, \qquad \mathscr{V}(\lambda; x, t) = \frac{2\mathrm{i}}{\lambda^2} M - \frac{1}{\lambda} M_x M + \mathrm{i}B. \tag{103}$$

The zero-curvature condition (102) is equivalent to the equation of motion (95). For $B = 0$, the latter is none other than conservation of the Noether current, namely the local continuity equation for matrix-valued charge density $M(x,t)$, $M_t + (\mathrm{i}[M, M_x]/2)_x = 0$.

**Example.** For a brief illustration, we consider the simplest example of a 2-sphere $\mathcal{M}_1 = S^2 \cong \mathbb{CP}^1$. As customary, we will represent the matrix field variable $M_\ell$ in terms of a unit vector (spin) field $\mathbf{S}_\ell \in S^2$ ($\mathbf{S}_\ell \cdot \mathbf{S}_\ell = 1$) in $\mathbb{R}^3$, $M_\ell = \mathbf{S}_\ell \cdot \boldsymbol{\sigma}$, where $\boldsymbol{\sigma} = (\sigma^{\mathrm{x}}, \sigma^{\mathrm{y}}, \sigma^{\mathrm{z}})^{\mathrm{T}}$ is a vector of Pauli matrices. The symplectic map $\Phi_\tau$ for this case has been studied previously in [24]

$$\Phi_\tau(\mathbf{S}_1, \mathbf{S}_2) = \frac{1}{\tau^2 + \varrho^2}\left(\varrho^2 \mathbf{S}_1 + \tau^2 \mathbf{S}_2 + \tau \mathbf{S}_1 \times \mathbf{S}_2, \varrho^2 \mathbf{S}_2 + \tau^2 \mathbf{S}_1 + \tau \mathbf{S}_2 \times \mathbf{S}_1\right), \tag{104}$$

where $\varrho^2 \equiv (1 + \mathbf{S}_1 \cdot \mathbf{S}_2)$.

To retrieve the semi-discrete and continuum limits of Eq. (104), we expand the inverses in Eq. (93), yielding

$$\frac{\mathrm{d}}{\mathrm{d}t}\mathbf{S}_\ell = \frac{\mathbf{S}_\ell \times \mathbf{S}_{\ell-1}}{1 + \mathbf{S}_\ell \cdot \mathbf{S}_{\ell-1}} + \frac{\mathbf{S}_\ell \times \mathbf{S}_{\ell+1}}{1 + \mathbf{S}_\ell \cdot \mathbf{S}_{\ell+1}} + \mathbf{S}_\ell \times \mathbf{B}, \tag{105}$$

where we have put $B = \mathbf{B} \cdot \boldsymbol{\sigma}/2$. We can recognize the integrable lattice discretization of the $SO(3)$-symmetric Heisenberg ferromagnet (isotropic lattice Landau–Lifshitz model [108,109]) in a homogeneous field $\mathbf{B}$. The equation of motion is generated by a logarithmic interaction of the form [18]

$$H_{\mathrm{LLL}} = -\sum_{\ell=1}^{L} \log(1 + \mathbf{S}_\ell \cdot \mathbf{S}_{\ell+1}) + \mathbf{B} \cdot \mathbf{S}_\ell. \tag{106}$$

Its long-wavelength limit yields Eq. (96), with the equation of motion

$$\mathbf{S}_t = \{\mathbf{S}, H_{\mathrm{c}}\} = -\mathbf{S} \times \frac{\delta H_{\mathrm{c}}}{\delta \mathbf{S}} = \mathbf{S} \times \mathbf{S}_{xx} + \mathbf{S} \times \mathbf{B}. \tag{107}$$

# 3 Charge transport and KPZ superuniversality

The remainder of the paper is devoted to a numerical study of equilibrium transport properties of integrable matrix models, with aim to address the central questions outlined in the introduction. To this end, we shall focus exclusively to transport of the Noether charges in canonical equilibrium ensembles where we can anticipate anomalous features. Here in particular we have in mind the previous studies of magnetization transport in the isotropic Landau–Lifshitz

(Heisenberg) magnet (with the spin-field belonging to the coset $G/H = S^2$) which uncovered superdiffusive transport of the KPZ universality class, both in the quantum and classical setting [24, 66–69, 76, 77]. The aim of the subsequent analysis is to systematically analyze the role of isometry and isotropy groups $G = SU(N)$ and $H = S(U(k) \times U(N-k))$, respectively. We shall also consider a distinct case of symplectic symmetry with $G = USp(2N)$ and $H = U(N)$.

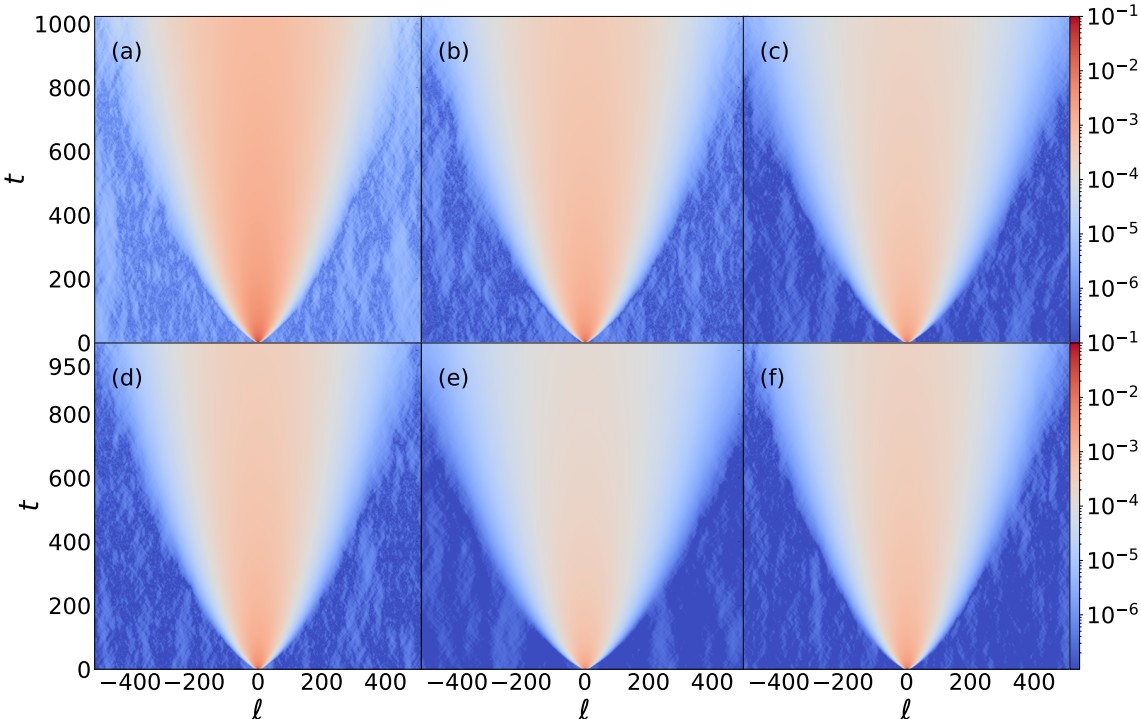

Figure 5: Space-time profiles of the charge autocorrelation function $C_{\mathbb{q}}(\ell, t)$ (shown the absolute value in logarithmic scale) for various local variables $M \in \mathcal{M}_1 = \mathrm{Gr}_{\mathbb{C}}(k, N)$: (a) $(k, N) = (1, 2)$, (b) $(k, N) = (1, 3)$, (c) $(k, N) = (1, 4)$, (d) $(k, N) = (2, 4)$, (e) $(k, N) = (1, 5)$ and (f) $(k, N) = (2, 5)$. The data shown for parameters $\tau = 1$, $N_{\mathrm{s}} = 10^5$ and $L = 2^{10}$.

The Noether charge represents a $G$-valued dynamical observable whose local densities are provided by the momentum map

$$f_X(M_\ell^t) = \mathrm{Tr}(X M_\ell^t). \tag{108}$$

We will subsequently use notation $q_\ell^a(t) \equiv f_{X^a}(M_\ell^t)$ for components of the Noether charge.

Exact computation of time-dependent correlation functions in equilibrium states lies beyond the capabilities of available analytic techniques. We thus have to fully rely on numerical simulations. The main object of study in our simulations are connected spatio-temporal autocorrelation functions of charge densities,

$$C_{q^a}(\ell, t) = \langle q_\ell^a(t) q_0^a(0) \rangle - \langle q_\ell^a(0) \rangle \langle q_0^a(0) \rangle. \tag{109}$$

Presently, the 'equilibrium expectation value' $\langle \cdot \rangle$ pertains to averaging with respect to a uniform Liouville measure on $\mathcal{M}_L$. The latter is an analogue of the canonical Gibbs state at 'infinite temperature' and is invariant under unit time and space shifts $t \to t+1$ and $\ell \to \ell+1$, respectively. Indeed, since $G$ acts transitively on $\mathrm{Gr}_{\mathbb{C}}(k, N)$, the $G$-invariant measure on Grassmannian manifolds is naturally inherited from the invariant (Haar) measure on $G$. In practice

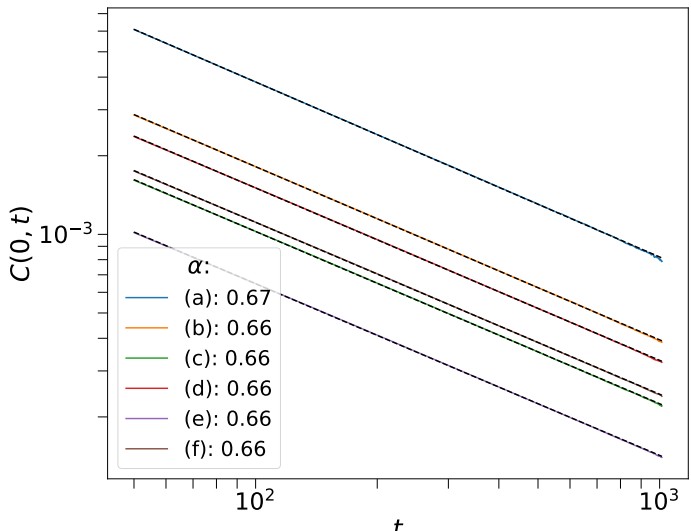

Figure 6: Algebraic dynamical exponents $\alpha = 1/z$ characterizing the asymptotic decay of correlators $C_{\mathbb{q}}(0,t) \sim |t|^{-\alpha}$ (for the corresponding datasets shown in Fig. 5) obtained by least square fit.

one can therefore first sample uniformly over the group $G$ (see e.g. [110]) and then generate the invariant distribution on $\mathrm{Gr}_{\mathbb{C}}(k,N)$ through the mapping $M = g \, \Sigma^{(k,N)} g^{\dagger}$, see Eq. (16).

We have numerically computed the dynamical correlator defined in Eq. (109) using the following scheme. First, we generated $N_{\mathrm{s}}$ initial matrix ensembles $\mathcal{E} \equiv \{M_{\ell}^{t=0}\}_{\alpha=1}^{N_{\mathrm{s}}}$ by drawing each sample set from the Liouville probability density $\rho^{(k,N)}$. Next, we computed the connected longitudinal dynamical correlators with the following prescription

$$\widehat{C}_{q^a}(x,t) = \frac{1}{N_{\mathrm{s}}} \sum_{\mathcal{E}} \frac{2}{(t_{\max}-t+1)N} \sum_{t'=0}^{t_{\max}-t} \sum_{\ell'=1}^{L/2} q_{\ell+2\ell'}^a(t+2t') q_{2\ell'}^a(2t') - \langle q^a \rangle^2, \qquad (110)$$

which can be efficiently performed using the convolution theorem. The maximal simulation time $t_{\max}$ can be adjusted so as to eliminate any spurious effect due to periodic boundary conditions.[12] To smear out the even-odd effect of staggering (see Figure 2), it is better to compute the autocorrelation function of the Noether charges by averaging over adjacent pairs of variables, $\mathbb{q}_{\ell} := \frac{1}{2}(q_{\ell} + q_{\ell+1})$. The corresponding 'smoothened' correlation function is given by

$$C_{\mathbb{q}}(\ell,t) = \langle \mathbb{q}_{\ell}(t) \mathbb{q}_0(0) \rangle - \langle \mathbb{q} \rangle^2 = \frac{1}{4}\widehat{C}(\ell-1,2t) + \frac{1}{2}\widehat{C}(\ell,2t) + \frac{1}{4}\widehat{C}(\ell+1,2t). \qquad (111)$$

Lastly, by virtue of the global $G$-invariance we are allowed to average over all the components $a = 1, 2, \ldots, \dim \mathfrak{g}$.

## 3.1 Uniform equilibrium states

In Figure 5 we display the time-dependent correlation functions (109) for the few smallest dimensions $N \in \{2, 3, 4, 5\}$ and all inequivalent signature specifications (i.e. $k = 1, 2, \ldots \lfloor N/2 \rfloor$). To study transport, twist fields must be set to identity, $F = \mathbb{1}$.

Assuming an *algebraic* decay at large times,

$$C_{\mathbb{q}}(\ell,t) \sim t^{-1/z} g\big((\lambda_{\mathrm{B}} t)^{-1/z} \ell\big), \qquad (112)$$

---

[12]Despite an extra sum over $t'$ in Eq. (110), there is no additional time averaging or any assumption of ergodicity involved; the purpose of this prescription is to extract the maximal amount of statistics from the data.

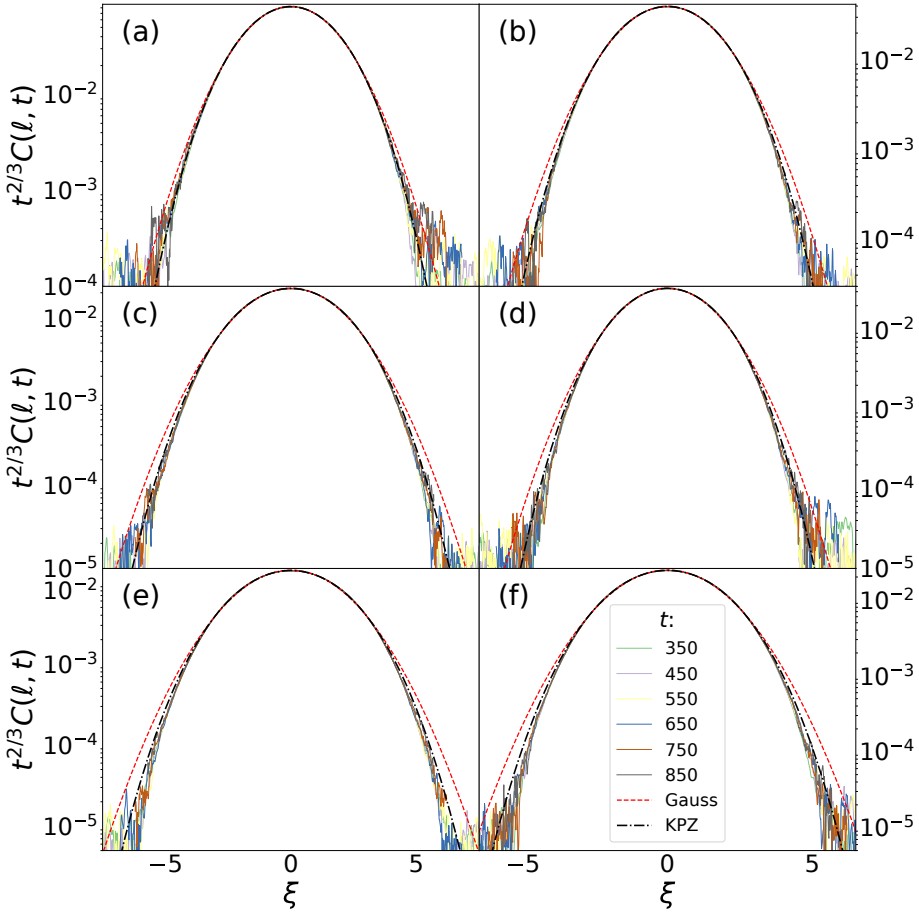

Figure 7: Convergence to the stationary cross sections of the scaled dynamical structure factors $\widetilde{C}_{\mathbb{q}}(\xi, t)$, fitted with the KPZ universal function $g_{PS}$ (black dashed curve), for the corresponding datasets shown in Fig. 5. In comparison, the red dashed lines display the best fit with a Gaussian profile (red dashed curve), showing systematic deviations in the tails.

we first extract the dynamical exponent $z$ from the numerical data. We find, uniformly for all the instances with $k \leq N/2$ and $N = 2, \ldots, 5$, excellent agreement with the Kardar–Parisi–Zhang superdiffusive universal algebraic exponent $z_{KPZ} = 3/2$, cf. Figure 6.

To further corroborate the presence of KPZ physics, we proceed with the extraction of the scaled dynamical structure factor

$$\widetilde{C}_{\mathbb{q}}(\xi, t) = t^{1/z} C_{\mathbb{q}}(\ell, t), \qquad \xi := \ell\, t^{-1/z}. \tag{113}$$

In Figure 7 we display the stationary cross sections which are expected to collapse onto a universal scaling function $g_{PS}$ tabulated in [111],

$$\lim_{t \to \infty} \widetilde{C}(\xi, t) = A\, g_{PS}\big(\lambda_B^{-1/z} \xi\big), \tag{114}$$

where $\lambda_B \in \mathbb{R}$ is the Burger's field coupling constant and A is the amplitude. Agreement with the universal KPZ profile is very solid and there are clearly visible systematic deviations from the Gaussian form which is characteristic of normal diffusion, see Figure 7. The non-Gaussian behaviour of the scaling function $g_{PS}$ is most pronounced in the tails, i.e. at large values of the scaling variable $\xi$. The extracted numerical values of constants $\lambda_B$ and A are reported in Table 1.

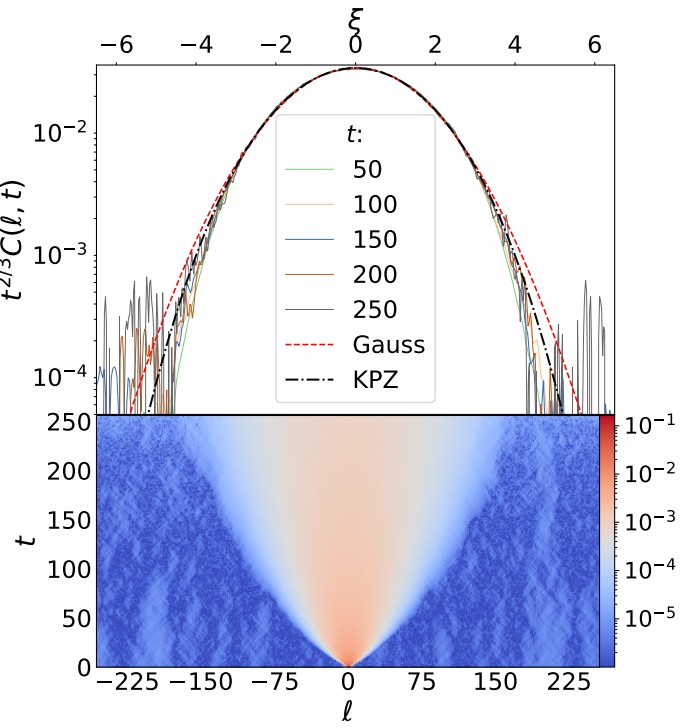

Figure 8: Dynamics of Noether charges in an invariant unbiased maximum-entropy state for the matrix model with $G = USp(4)$ and the Lagrangian Grassmannian L(2) as the local phase-space manifold, showing relaxation of the charge correlator (bottom panel) and convergence towards stationary KPZ scaling profile (top panel).

For completeness we include the numerical analysis of charge transport in an integrable matrix model on a Lagrangian Grassmannian. Since these are sub-manifolds of complex Grassmannians, they have to be considered independently. We shall only consider here the simplest instance $L(2) \cong USp(2; \mathbb{C})/U(2)$. Numerical data shown in Figure 8 again complies well with the KPZ scaling despite of having a different symmetry type.

### 3.1.1 Magnetic field

To incorporate an external $SU(N)$-magnetic field we set the twisting element to

$$F = \exp(-\mathrm{i}\,\tau B/2), \tag{115}$$

where $B$ is a fixed Hermitian matrix of the form $B = h \sum_a n_a X^a$, with field strength $h$ and (unit) polarization vector $\mathbf{n}$. The addition of a field causes the following dynamical effect: the dynamical correlations pertaining to the distinguished Noether charge aligned with the polarization direction is unaffected by the field, whereas the correlators of all the remaining charges exhibit a super-diffusive KPZ spreading modulated by a periodic precessional motion, see Figure 9.

### 3.1.2 Inhomogeneous phase space

We finally explore an interesting possibility of introducing an integrable matrix model on an inhomogeneous phase space of the form

$$\mathcal{M}_L^{(\mathbf{k},N)} \equiv \mathcal{M}_1^{(k_1,N)} \times \mathcal{M}_1^{(k_2,N)} \times \cdots \times \mathcal{M}_1^{(k_L,N)}. \tag{116}$$

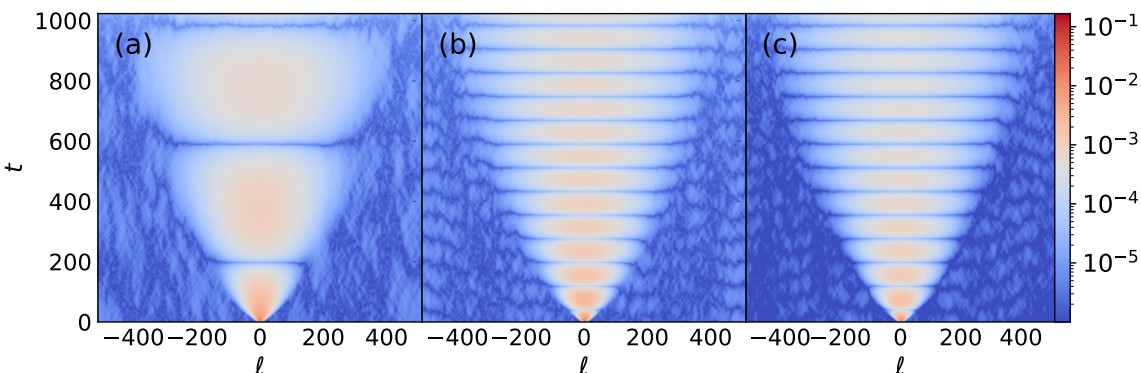

Figure 9: Effect of an applied magnetic field $F_\tau = \exp(-i\tau(h/2)\sum_a n_a X^a)$ (cf. Eq. (115)) on the correlation function of the Noether charges perpendicular to the polarization direction **n**, shown for (a) $N = 2$, $h = 10^{-3}$, (b) $N = 2$, $h = 10^{-2}$, (c) $N = 3$, $h = 10^{-2}$ (with parameters $N_s = 10^3$, and $L = 2^{10}$).

| $(N, k)$ | $\lambda_B \times 10^2$ | $A_f \times 10^2$ | $A_i \times 10^2$ |
|---|---|---|---|
| $(2, 1)$ | 8.41 | 8.29 | 8.38 |
| $(3, 1)$ | 7.45 | 3.99 | 3.85 |
| $(4, 1)$ | 6.33 | 2.27 | 2.14 |
| $(4, 2)$ | 8.44 | 3.33 | 3.14 |
| $(5, 1)$ | 5.42 | 1.44 | 1.33 |
| $(5, 2)$ | 8.13 | 2.47 | 2.30 |
| $L(2)$ | 9.18 | 3.43 | 3.31 |

Table 1: Numerical values of the Burger's coupling constant $\lambda_B$ (profile width) and amplitude (height) A, characterizing stationary KPZ profiles computed for several lowest dimensions $N$ and ranks $k$. Parameters $(\lambda_B, A_f)$ were obtained by fitting the scaling function (114). Amplitudes $A_i$ were read off from the horizontal axis intercepts of the equal-space correlator $C(0, t)$ shown in Figure 6. The last line pertains to the unitary symplectic case (see Figure 8).

Such a staggered structure is still compatible with integrability of the many-body dynamics. This is a corollary of the fact that $\Phi_\tau$ acts as a conjugation in $G \times G$ which preserves the the total signature by swapping the signature of two adjacent incidence matrices $M$ and $M'$, allowing to 'scatter' degrees of freedom from different adjoint orbits. As a consequence, the total signature $\sum_\ell \Sigma^{(k_\ell, N)}$ is conserved under time evolution.

**Staggered phase space.** As an illustration of the above construction we consider a special case of a staggered phase space with an alternating sequence of inequivalent phase spaces of rank $k = 1$ and $k' = 2$, specializing to the lowest-dimensional instance $N = 4$. As shown in Figure 10, staggering induces a chiral structure in the problem causing an asymmetric spreading of correlations. The dynamical correlations of Noether charges experience a linear drift, combined with superdiffusive spreading with a dynamical exponent indistinguishable from $z_{KPZ} = 3/2$. This time, however, stationary profiles do not appear to converge towards the KPZ scaling function. In fact, we find an asymmetric profile with discernible deviations in the left tail which seem unrelated to finite-time effects.

Owing to an intrinsic chiral structure of this model, we have also tried a two-sided fit by fitting the KPZ scaling function for each chiral component (left and right movers) separately. Doing this however did not appreciably improve upon the fit in Figure 10. We postpone a more

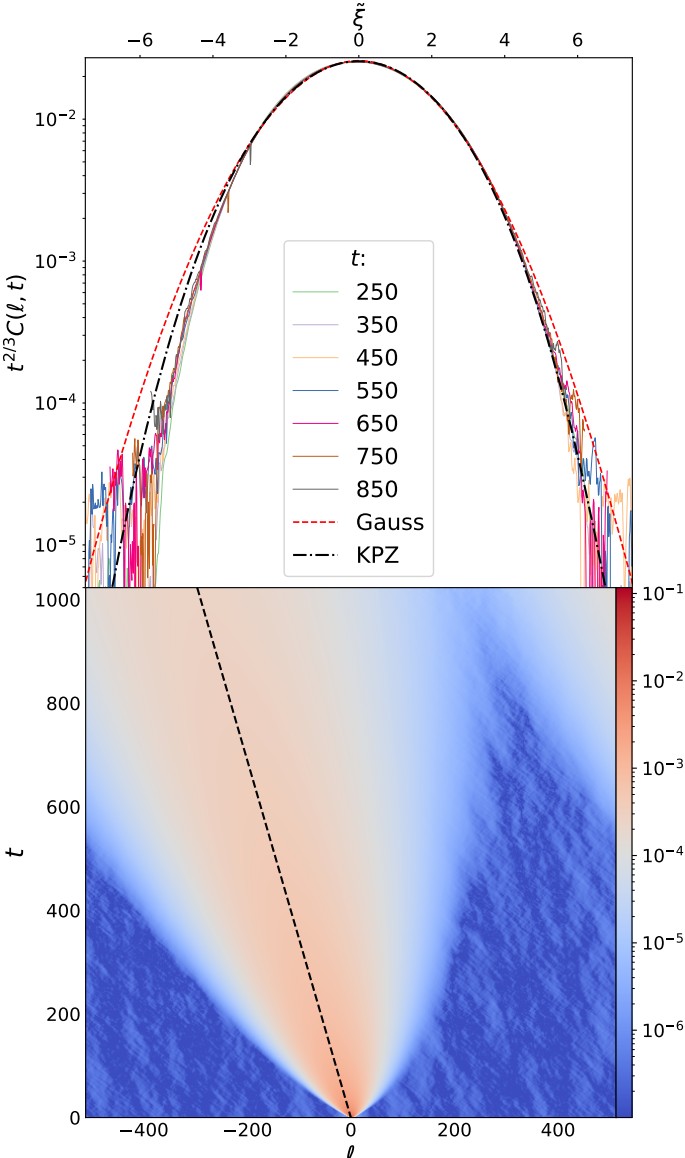

Figure 10: Evolution of the dynamical charge correlation function in a matrix model with an inhomogeneous phase space of the staggered type, shown for the simplest case of complex Grassmannians $\mathrm{Gr}_{\mathbb{C}}(1,4)$ and $\mathrm{Gr}_{\mathbb{C}}(2,4)$ (bottom panel). Black dotted line marks the center of the correlation. Scaled correlator with dynamical exponent $\alpha = 2/3$ characteristic of KPZ superdiffusion (top panel). The scaling variable is defined as $\tilde{\xi} = (\ell + vt)/t^{2/3}$, with $v = 0.29$.

detailed analysis of this exceptional scenario for future work.

## 4  Discussion and conclusion

We have introduced a novel family of classical integrable models of interacting matrix-valued degrees of freedom propagating on a discrete space-time lattice, and obtained an explicit dynamical system in the form of a classical Floquet circuit composed of elementary two-body symplectic maps. The class of models is distinguished by the presence of a conserved $G$-invariant Noether currents and (in general non-Abelian) local gauge invariance under a subgroup $H$. In

the absence of external fields, both time and space dynamics can be realized in uniform way, which reveals a particular type of space-time self-duality. Integrability of our models manifest itself through the discrete zero-curvature condition on the light-cone lattice, implying infinitely many conserved quantities in involution and a set-theoretic Yang–Baxter relation for the elementary two-body symplectic propagator.

Integrable difference equations have been extensively studied in the mathematical physics literature (see [81,82]), particularly in the context of equations on quadrilateral graphs [101, 112, 113] which have been classified in the work of Adler, Bobenko and Suris [114]. These include, as a prominent example, the Faddeev–Volkov discretization of the sine-Gordon model [115]. To our knowledge, the class of models introduced in this work does is not a part of any known classification scheme, despite from the viewpoint of conceptual simplicity they can hardly be rivalled. An alternative, albeit less explored, approach to produce integrable difference equations has been developed in [116–118] and [119–121], via discretization of Hirota derivatives [122,123]. Although in Ref. [119] the authors obtained a particular lattice discretization of the $N = 2$ isotropic Landau–Lifshitz model, its implicit form makes it less appealing for concrete applications. In more recent works [103,104], several formal connections between quantum Yang–Baxter maps (representing the adjoint action of the universal $R$-matrix of a quantum group) and their classical limits (and discrete-time dynamics) have been uncovered, indicating that classical Yang–Baxter maps in a way naturally descend from the associated quantized algebraic structure. In this respect, our results indicate that, at the set-theoretic level, the emergent classical Yang–Baxter map does not show explicit dependence on the underlying Lie algebra.

In the second part of the paper, the outlined explicit integration scheme have been employed as an efficient numerical tool to investigate transport properties of the Noether charges in unbiased maximum-entropy states. In close analogy to the isotropic Landau–Lifshitz model, we now found robust evidence of the superdiffusive transport in the KPZ universality class, irrespectively of the structure of the local phase space, i.e. isometry and isotropy groups of their coset target spaces. In fact, universality of KPZ type extends even to Lagrangian Grassmannians that are linked with unitary symplectic groups. While the outlined construction does not accommodate for matrix models associated with real orthogonal Lie groups or exceptional compact groups, we expect that these could be included with suitable adaptations.

We nonetheless believe that the following conjecture can be stated: *all discrete space-time models built as Floquet circuits from two-body symplectic Yang-Baxter maps (and continuum limits thereof), with dynamical variables taking values on compact non-abelian symmetric spaces, exhibit superdiffusion of the KPZ type in equilibrium states with unbroken symmetry*. If the conjecture holds, the observed phenomenon of KPZ physics can be dubbed as *superuniversal*. Perhaps this conjecture could be even slightly expanded by adjoining matrix models on supersymmetric coset spaces invariant under Lie superalgebras (it is known from [37] that the corresponding integrable quantum chains possess divergent diffusion constants). We postpone further examination and other related questions to future studies, including a more comprehensive study of charge transport by extending the analysis to grand-canonical states.

Our models provide integrable Trotterizations of (non-relativistic) coset $\sigma$-models on complex Grassmannian manifolds, emerging as semi-classical limits of integrable symmetric quantum spin chains invariant under global $SU(N)$ rotations that are known to exhibit anomalous charge transport [37]. It thus appears plausible that our findings elevate to the quantum setting too. This would suggest that there is a general principle behind an exact quantum–classical correspondence of charge transport, as exemplified previously in the scope of the domain wall problem in the Heisenberg model [23,124].

There are several distinct features of our models that were left unexplored but definitely merit further study. On the formal side, developing a fully-fledged inverse scattering formalism to integrate the auxiliary linear problem in discrete space-time would provide a platform to tackle various problems of nonequilibrium statistical mechanics in an analytical fashion. Another pending question is the fate of local conserved quantities outside of the 'projective models' ($k = 1$). Since for $\mathrm{Gr}_{\mathbb{C}}(k \geq 2, N)$ the monodromy matrices evaluated at the projection points no longer decompose into a sequence of rank-1 projectors, it is not obvious which mechanism (if any) would ensure locality of conserved quantities. Curiously however, in the limit of continuous time the symplectic generator of the elementary propagator yields a strictly local Hamiltonian density that generates the time evolution of their integrable lattice counterparts even for generic (i.e. non-projective) models ($k \geq 2$). It is not inconceivable that these lattice models involve quasi-local conservation laws, bearing some resemblance to higher-spin commuting transfer matrices in quantum Heisenberg model where the 'shift point' property also ceases to exist [125, 126]. Another interesting question which remains open is whether it is possible to take different continuum limits to systematically recover the entire hierarchy of higher Hamiltonian flows in the field-theory limit.

Another curiosity of our models is their space-time self-duality property. While the latter formally permits to study dynamics in the space direction, that is evolutions of time-states (see e.g. [28]), it remains obscure at this moment if this has any implications on the structure of dynamical correlations in these matrix models, or possibly even for the observed anomalous transport behavior. We wish to stress here that this type of self-duality differs fundamentally from the so-called dual-unitarity found recently in the context of quantum circuits [99, 127] where correlations are locked to the light-rays. In our models, dynamical correlations functions (averaged in a flat invariant measure) fill the entire causal cone in a non-trivial fashion.

To facilitate other physical applications, it would be valuable to obtain various integrable deformations such as adding an uniaxial interaction anisotropy. Based on numerical evidence from the anisotropic Landau–Lifshitz model [76], spin transport depends quite intricately on the value of anisotropy, ultimately responsible for the elusive behavior of the gapless phase in the anisoropic Heisenberg XXZ spin-1/2 chain [128, 129] (see [126] for a review). To conclude, we wish to mention that the observed superuniversal nature of the KPZ phenomenon in integrable models with non-abelian Noether currents gives a further (albeit implicit) hint that the notion of hydrodynamic soft modes, employed recently in a phenomenological description of the KPZ phenomenon [41, 42], extends beyond the simplest $SO(3)$-invariant Landau–Lifshitz theory. We postpone the study of these aspects for the future.

## Acknowledgements

We thank O. Gamayun and P. Saksida for their remarks. TP acknowledges stimulating related discussions with V. Pasquier in the preliminary stage of this project. The work has been supported by ERC Advanced grant 694544 – OMNES and the program P1-0402 of Slovenian Research Agency.

# Appendices

## A  Two-body propagator

Here we show that the discrete zero curvature condition

$$FL(\lambda; M_2)L(\mu; M_1) = L(\mu; M_2')L(\lambda; M_1')F, \tag{117}$$

with a linear Lax matrix $L(\lambda; M) = \lambda \mathbb{1} + iM$ admits a *unique* solution $\Phi(\lambda, \mu) : (M_1, M_2) \mapsto (M_1', M_2')$ provided that

1. matrix variables $M$ obey the involutory constraint $M^2 = \mathbb{1}$,

2. $\Phi(\lambda, \mu)$ is of the difference form, i.e. it depends only on $\tau := \mu - \lambda$, where $\tau$ is a fixed (generally complex) parameter.

Equation (117) can be regarded as an identity for two quadratic polynomials in the spectral parameter $\lambda \in \mathbb{C}$. We must thus equate each power in $\lambda$. Expanding out the zero-curvature condition (117) and dropping the leading $\lambda^2$ terms, we find

$$\lambda(iM_1 + iM_2 + \tau) + (i\tau M_2 - M_2 M_1) = \mathrm{Ad}_{F^{-1}}\Big(\lambda(iM_1' + iM_2' + \tau) + (i\tau M_1' - M_2' M_1')\Big). \tag{118}$$

Matching the linear terms in $\lambda$ immediately implies a two-site conservation law (6)

$$M_1' + M_2' = F(M_1 + M_2)F^{-1} \equiv \mathrm{Ad}_F(M_1 + M_2), \tag{119}$$

while the $\lambda^0$ term gives

$$-i\tau M_2 + M_2 M_1 = \mathrm{Ad}_{F^{-1}}\left(-i\tau M_1' + M_2' M_1'\right). \tag{120}$$

Using Eq. (119), we can bring it to the form

$$M_1 M_2 (M_1 + M_2 - i\tau) = F^{-1}(M_1' + M_2' - i\tau)M_2' M_1' F, \tag{121}$$

from where it follows

$$M_2' M_1' = F(M_1 + M_2 - i\tau)^{-1} M_1 M_2 (M_1 + M_2 - i\tau)F^{-1}. \tag{122}$$

Plugging the above back into Eq. (120) to eliminate the product of primed variables, and writing $S_\tau = M_1 + M_2 + i\tau$, we arrive at the following explicit form

$$M_1' = \mathrm{Ad}_F\Big(M_2 + \frac{i}{\tau}M_2 M_1 - \frac{i}{\tau}(S_{-\tau})^{-1} M_1 M_2 S_{-\tau}\Big), \tag{123}$$

$$M_2' = \mathrm{Ad}_F\Big(M_1 - \frac{i}{\tau}M_2 M_1 + \frac{i}{\tau}(S_{-\tau})^{-1} M_1 M_2 S_{-\tau}\Big). \tag{124}$$

In this way, we have established uniqueness of the time propagator $\Phi_\tau$. The right-hand side can be brought into the final form (8) by a straightforward computation.

**Closure on $\mathrm{Gr}_{\mathbb{C}}(k,N)$.** It can now be readily demonstrated that the mapping (8) preserves Grassmannian submanifolds $\mathrm{Gr}_{\mathbb{C}}(k,N) = SU(N)/S(U(k) \times U(N-k))$ of the unitary group $SU(N)$ if $\tau \in \mathbb{R}$, without explicitly specifying $N$ and $k$.

Considering two hermitian matrices $M_{1,2}$ and a unitary twisting matrix $F$, the hermitian-conjugate counterpart of

$$M_1' = FS_\tau M_2 (FS_\tau)^{-1}, \tag{125}$$

reads

$$(M_1')^\dagger = F(S_{-\tau})^{-1} M_2 S_{-\tau} F^{-1}. \tag{126}$$

To establish $M_1' = (M_1')^\dagger$, is remains to verify explicitly that $S_{-\tau}S_\tau M_2 = M_2 S_{-\tau} S_\tau$, which is a matter of a short calculation. Preservation of the involutory property under $\Phi_\tau$ is manifest from Eq. (125).

**Closure on $\mathrm{L}(N)$.** Let $M_{1,2}$ now be two unitary *antisymplectic* complex matrices of dimension $2N$ with the involutory property, and $F$ a symplectic or antisymplectic matrix, namely

$$M_{1,2}M_{1,2}^\dagger = (M_{1,2})^2 = \mathbb{1}, \qquad M_{1,2}^{\mathrm{T}} J M_{1,2} = -J, \qquad F^{\mathrm{T}} J F = \pm J, \tag{127}$$

with $J \equiv \mathrm{i}\,\sigma^y \otimes \mathbb{1}_N$. The involutory property implies $M_{1,2}^{\mathrm{T}} J = -J M_{1,2}$. It can be checked directly that $M_{1,2}'$ retain the antisymplectic property. In contrast, another short calculation shows that the symplectic property of a pair $(M_1, M_2)$ is not preserved. The dynamical map (8) is thus closed on the Lagrangian Grassmannian manifold $\mathrm{L}(N) = USp(2N)/U(N)$.

## A.1 Uniqueness of factorization

The proof of the set-theoretic Yang-Baxter relation (86), shortly outlined in Section 2.4.3, hinges on uniqueness of factorization of an ordered product of Lax matrices $L(\lambda_j; M_j)$, with parameters $\lambda_j = \lambda + \zeta_j$ and $\lambda, \zeta_j \in \mathbb{C}$. The vital part of the proof is to establish that equivalence of the following two cubic monodromies,

$$L(\lambda_3; M_3^\circ)L(\lambda_2; M_2^\circ)L(\lambda_1; M_1^\circ) = L(\lambda_3; M_3^\bullet)L(\lambda_2; M_2^\bullet)L(\lambda_1; M_1^\bullet), \tag{128}$$

implies that all the matrix variables are pairwise equal, that is $M_j^\circ = M_j^\bullet$ for all $j \in \{1,2,3\}$. This assertion can be proven with a bit of algebraic manipulations. To this end, it is convenient to use a slightly adapted notation and introduce matrices $Y_j^\circ = \mathrm{i} M_j^\circ + (\zeta_j - \zeta_3)\mathbb{1}$ (and analogously for $Y_j^\bullet$ variables)[13] in terms of which

$$L(\lambda; M_j) = \lambda_3 \mathbb{1} + Y_j. \tag{129}$$

Furthermore, by redefining the spectral parameter as $\lambda \to \lambda - \zeta_3$ and expanding Eq. (128) in powers $\lambda$, we obtain the following systems of matrix equations

$$Y_3^\circ + Y_2^\circ + Y_1^\circ = Y_3^\bullet + Y_2^\bullet + Y_1^\bullet, \tag{130}$$

$$Y_3^\circ Y_2^\circ + Y_3^\circ Y_1^\circ + Y_2^\circ Y_1^\circ = Y_3^\bullet Y_2^\bullet + Y_3^\bullet Y_1^\bullet + Y_2^\bullet Y_1^\bullet, \tag{131}$$

$$Y_3^\circ Y_2^\circ Y_1^\circ = Y_3^\bullet Y_2^\bullet Y_1^\bullet. \tag{132}$$

Rewriting Eq. (132) as

$$Y_3^\circ Y_2^\circ Y_1^\circ + Y_3^\circ Y_3^\circ Y_2^\circ + Y_3^\circ Y_3^\circ Y_1^\circ - Y_3^\circ + Y_3^\circ + Y_2^\circ + Y_1^\circ =$$
$$Y_3^\bullet Y_2^\bullet Y_1^\bullet + Y_3^\bullet Y_3^\bullet Y_2^\bullet + Y_3^\bullet Y_3^\bullet Y_1^\bullet - Y_3^\bullet + Y_3^\bullet + Y_2^\bullet + Y_1^\bullet, \tag{133}$$

---

[13]Notice that variables $Y_j$ are no longer anti-involutory, with the exception of $Y_3$.

subtracting from it Eq. (130), and finally factoring out $Y_3$ from the left, we find

$$Y_3^\circ(Y_2^\circ Y_1^\circ + Y_3^\circ Y_2^\circ + Y_3^\circ Y_1^\circ - \mathbb{1}) = Y_3^\bullet(Y_2^\bullet Y_1^\bullet + Y_3^\bullet Y_2^\bullet + Y_3^\bullet Y_1^\bullet - \mathbb{1}). \tag{134}$$

Since both terms in the brackets are equal as per Eq. (131), assuming that they are generically non-vanishing, we deduce

$$Y_3^\circ = Y_3^\bullet. \tag{135}$$

Having shown this, we can simply act on Eq. (128) from the left by an inverse of the Lax matrix and thus reduce the problem to that of a two-body factorization. The latter can be then solved in a manner analogous to the above procedure. In this way we have established that $M_j^\circ = M_j^\bullet$ is the unique solution to Eq. (88). As a matter of fact, using the same logic (shifting the spectral parameter, introducing the $Y$-variables, expanding in $\lambda$, adding and subtracting terms using the conservation law at the order $\lambda^0$ to factor out the remaining anti-involutory variable $Y_L$) one can establish uniqueness of factorization for an arbitrary monodromy matrix of length $L$.

## B  Symplectic properties

Here we present a direct proof that Eq. (8) provides a symplectic map (i.e. symplectomorphism) on the product of two Grassmannians manifolds $\mathcal{M}_1 \times \mathcal{M}_1$ and, as a consequence, conserves the product Liouville measure. To establish this, it suffices to demonstrate that the Poisson bracket is preserved under the time-evolution. It proves convenient to carry out this computation using the Sklyanin bracket.

We first treat the bracket involving a pair of variables on the same space $\mathcal{M}_1$. Separating out the twist dependence,

$$\{M_2' \overset{\otimes}{,} M_2'\} = \mathrm{Ad}_{F \otimes F}\{S_\tau M_1 S_\tau^{-1} \overset{\otimes}{,} S_\tau M_1 S_\tau^{-1}\}, \tag{136}$$

and subsequently expanding everything out using the Leibniz derivation rule, a tedious calculation yields

$$\begin{aligned}
\{S_\tau M_1 S_\tau^{-1} \overset{\otimes}{,} S_\tau M_1 S_\tau^{-1}\} = {} & (S_\tau M_1 S_\tau^{-1} \otimes S_\tau M_1 S_\tau^{-1})\{S_\tau \overset{\otimes}{,} S_\tau\}(S_\tau^{-1} \otimes S_\tau^{-1}) \\
& + \{S_\tau \overset{\otimes}{,} S_\tau\}(M_1 S_\tau^{-1} \otimes M_1 S_\tau^{-1}) + (S_\tau \otimes S_\tau)\{M_1 \overset{\otimes}{,} M_1\}(S_\tau^{-1} \otimes S_\tau^{-1}) \\
& + (S_\tau \otimes \mathbb{1})\{M_1 \overset{\otimes}{,} M_1\}(S_\tau^{-1} \otimes M_1 S_\tau^{-1}) \\
& + (\mathbb{1} \otimes S_\tau)\{M_1 \overset{\otimes}{,} M_1\}(M_1 S_\tau^{-1} \otimes S_\tau^{-1}) \\
& - (S_\tau M_1 S_\tau^{-1} \otimes \mathbb{1})\{S_\tau \overset{\otimes}{,} S_\tau\}(S_\tau^{-1} \otimes M_1 S_\tau^{-1}) \\
& - (\mathbb{1} \otimes S_\tau M_1 S_\tau^{-1})\{S_\tau \overset{\otimes}{,} S_\tau\}(M_1 S_\tau^{-1} \otimes S_\tau^{-1}) \\
& - (S_\tau M_1 S_\tau^{-1} \otimes S_\tau)\{M_1 \overset{\otimes}{,} M_1\}(S_\tau^{-1} \otimes S_\tau^{-1}) \\
& - (S_\tau \otimes S_\tau M_1 S_\tau^{-1})\{M_1 \overset{\otimes}{,} M_1\}(S_\tau^{-1} \otimes S_\tau^{-1}). \tag{137}
\end{aligned}$$

The obtained expression be further simplified with aid of general matrix identities,

$$(A \otimes B)\{M_{1,2} \overset{\otimes}{,} M_{1,2}\}(C \otimes D) = \mathrm{i}\big((AM_{1,2}D \otimes BC) - (AD \otimes BM_{1,2}C)\big)\Pi, \tag{138}$$

$$(A \otimes B)\{S_\tau \overset{\otimes}{,} S_\tau\}(C \otimes D) = \mathrm{i}\big((AS_\tau D \otimes BC) - (AD \otimes BS_\tau C)\big)\Pi, \tag{139}$$

which hold for any set of dummy $N$-dimensional matrices $\{A, B, C, D\}$.[14] Using these, Eq. (137) can be brought into the form

$$\{M_2 \overset{\otimes}{,} M_2\} = \mathrm{i}\big(M_2' \otimes \mathbb{1} - \mathbb{1} \otimes M_2'\big)\Pi = -\frac{\mathrm{i}}{2}\big[\Pi, M_2' \otimes \mathbb{1} - \mathbb{1} \otimes M_2'\big], \tag{140}$$

---

[14]Note that to arrive at Eq. (139), the specific form of $S_\tau$ has to be taken into account.

in agreement with Eq. (45). To establish ultra-locality of the Poisson bracket (cf. Eq. (46)) we need to additionally verify that

$$\{M_1' \overset{\otimes}{,} M_2'\} = \mathrm{Ad}_{F \otimes F} \{S_\tau M_2 S_\tau^{-1}, S_\tau M_1 S_\tau^{-1}\} = 0. \tag{141}$$

This can once again be confirmed with an explicit but lengthy computation,

$$
\begin{aligned}
\{S_\tau M_2 S_\tau^{-1} \overset{\otimes}{,} S_\tau M_1 S_\tau^{-1}\} = {} & (S_\tau M_2 S_\tau^{-1} \otimes S_\tau M_1 S_\tau^{-1}) \{S \overset{\otimes}{,} S\} (S_\tau^{-1} \otimes S_\tau^{-1}) \\
& + \{S_\tau \overset{\otimes}{,} S_\tau\} (M_2 S^{-1} \tau \otimes M_1 S_\tau^{-1}) \\
& + (S_\tau \otimes \mathbb{1}) \{M_2 \overset{\otimes}{,} M_2\} (S_\tau^{-1} \otimes M_1 S_\tau^{-1}) \\
& + (\mathbb{1} \otimes S_\tau) \{M_1 \overset{\otimes}{,} M_1\} (M_2 S_\tau^{-1} \otimes S_\tau^{-1}) \\
& - (S_\tau M_2 S_\tau^{-1} \otimes \mathbb{1}) \{S_\tau \overset{\otimes}{,} S_\tau\} (S_\tau^{-1} \otimes M_1 S_\tau^{-1}) \\
& - (\mathbb{1} \otimes S_\tau M_1 S_\tau^{-1}) \{S_\tau \overset{\otimes}{,} S_\tau\} (M_2 S_\tau^{-1} \otimes S_\tau^{-1}) \\
& - (S_\tau M_2 S_\tau^{-1} \otimes S_\tau) \{M_1 \overset{\otimes}{,} M_1\} (S_\tau^{-1} \otimes S_\tau^{-1}) \\
& - (S_\tau \otimes S_\tau M_1 S_\tau^{-1}) \{M_2 \overset{\otimes}{,} M_2\} (S_\tau^{-1} \otimes S_\tau^{-1}) \Big),
\end{aligned}
\tag{142}
$$

which can be eventually simplified to zero.

## B.1 Symplectic generator

The aim of this section is to reformulate the two-body symplectic map (7) as a Hamiltonian equation of motion of the form

$$\frac{\mathrm{d}}{\mathrm{d}t} M_{1,2} = \{M_{1,2}, \mathcal{H}_\tau^{(k,N)}\}, \tag{143}$$

such that at time $t = \tau$ the continuous–time evolution generated by $\mathcal{H}_\tau^{(k,N)}$ matches the two-body time-propagator (7). The exact form of the symplectic generator depends on the matrix dimension $N$, rank $k$ and the time-step parameter $\tau$.

For simplicity we shall assume here the absence of a magnetic field, i.e. set the twist to $F = \mathbb{1}$. The generator $\mathcal{H}_\tau^{(k,N)}$ can be sought as a function

$$\mathcal{H}_\tau^{(k,N)} = \mathcal{H}_\tau^{(k,N)}(s_2, s_4, \dots), \tag{144}$$

of the scalar invariants of $S_0 \equiv M_1 + M_2$, namely

$$s_m = \mathrm{Tr}(S_0^m) \equiv \mathrm{Tr}\big((M_1 + M_2)^m\big), \qquad m \in \mathbb{N}. \tag{145}$$

These can alternatively be expressed as linear combinations of real invariants of hermitian involutory matrices of the form $\mathrm{Tr}(M_1 M_2 M_1 M_2 \dots M_1 M_2)$. It is not difficult to recognize that traces of odd powers are all proportional to $s_1 = \mathrm{Tr}(M_1 + M_2) = 2(N - 2k)$,

$$s_{2m+1} = 4^m s_1, \qquad m \in \mathbb{Z}_{\geq 0}. \tag{146}$$

It is thus only the even invariants $s_{2k}$ that are non-trivial functions of the dynamical variables.

By application of Leibniz's rule, the equations of motion are therefore put in the form

$$\frac{\mathrm{d}M_{1,2}}{\mathrm{d}t} = \sum_{k=0}^{\infty} \frac{\partial \mathcal{H}_\tau^{(k,N)}}{\partial s_{2k}} \{M_{1,2}, s_{2k}\}, \tag{147}$$

where for the time being no upper limit in the summation has been imposed. The vital part of the derivation is to compute the Poisson brackets $\{M_{1,2}, s_{2k}\}$, which can be achieved by using partial traces. Exploiting a useful identity,

$$\mathrm{Tr}_2\Big(\big[\Pi, A \otimes \mathbb{1} - \mathbb{1} \otimes A\big](\mathbb{1} \otimes B)\Big) = -2[A, B], \tag{148}$$

we proceed by evaluating the Poisson bracket

$$\begin{aligned}
\{M_{1,2}, s_m\} &= \mathrm{Tr}_2\big(\{M_{1,2} \overset{\otimes}{,} (M_1 + M_2)^m\}\big) \\
&= -\frac{\mathrm{i}m}{2}\mathrm{Tr}_2\Big((\mathbb{1} \otimes (M_1 + M_2)^{m-1})\big[\Pi, M_{1,2} \otimes \mathbb{1} - \mathbb{1} \otimes M_{1,2}\big]\Big) \\
&= \mathrm{i}m\big[M_{1,2}, (M_1 + M_2)^{m-1}\big].
\end{aligned} \tag{149}$$

This readily allows us to deduce the equations of motion

$$\frac{\mathrm{d}M_{1,2}}{\mathrm{d}t} = \mathrm{i}\sum_{m=1}^{\infty} 2m\frac{\partial \mathscr{H}_\tau^{(k,N)}}{\partial s_{2m}}\big[M_{1,2}, (M_1 + M_2)^{2m-1}\big]. \tag{150}$$

From this expression it is manifest that the sum $M_1 + M_2$ is conserved under the time evolution, permitting us to write the Heisenberg equation of motion

$$\frac{\mathrm{d}}{\mathrm{d}t}M_{1,2}(t) = \mathrm{i}[M_{1,2}(t), \mathscr{F}_\tau^{(k,N)}], \qquad \mathscr{F}_\tau^{(k,N)} \equiv \sum_{m=1}^{\infty} 2m\frac{\partial \mathscr{H}_\tau^{(k,N)}}{\partial s_{2m}}(M_1 + M_2)^{2m-1}, \tag{151}$$

with the solution

$$M_{1,2}(t) = \exp\big(-\mathrm{i}\,t\,\mathscr{F}_\tau^{(k,N)}\big)M_{1,2}(0)\exp\big(\mathrm{i}\,t\,\mathscr{F}_\tau^{(k,N)}\big). \tag{152}$$

Noticing another useful property of the map (7) at $t = 0$,

$$M_{1,2} = S_0^{-1}M_{2,1}S_0, \tag{153}$$

the solution can be cast in the form

$$M_{1,2}(t) = \exp\big(-\mathrm{i}\,t\,\mathscr{F}_\tau^{(k,N)}\big)S_0 M_{2,1}(0)S_0^{-1}\exp\big(\mathrm{i}\,t\,\mathscr{F}_\tau^{(k,N)}\big). \tag{154}$$

Evaluating now the solution at $t = \tau$ we can deduce a neat time-translation property,

$$S_\tau = \exp\big(-\mathrm{i}\,\tau\,\mathscr{F}_\tau^{(k,N)}\big)S_0, \tag{155}$$

which rewards us with a remarkably simple expression for $\mathscr{F}_\tau^{(k,N)}$,

$$\mathscr{F}_\tau^{(k,N)} = \frac{\mathrm{i}}{\tau}\log\big(\mathbb{1} + \mathrm{i}\tau S_0^{-1}\big). \tag{156}$$

By expanding the logarithm into a Taylor series, multiplying by $S_0^m$ and taking the trace, we find an infinite system of linear partial differential equations

$$\sum_{m=1}^{\infty} 2m\frac{\partial \mathscr{H}_\tau^{(k,N)}}{\partial s_{2m}}s_{2m+n-1} = \frac{\mathrm{i}}{\tau}\sum_{j=1}^{\infty}\frac{(-\mathrm{i}\tau)^j}{j}s_{n-j}, \qquad n \in \mathbb{Z}. \tag{157}$$

There are sufficiently many independent equations to ensure the solution to this system. This provides us with the symplectic generator $\mathscr{H}_\tau^{(k,N)}$, uniquely up to additive constants. Indeed, the solution is guaranteed to exist by symplecticity of the time-propagator $\Phi_\tau$.

The infinite system (157) can be further reduced to a finite closed system of differential equations by performing a resummation of the matrix invariants $s_k$. This can achieved by means of the Cayley–Hamilton theorem which states that every $N$-dimensional matrix $A$ satisfies its own characteristic polynomial,

$$p(\xi) = \text{Det}(\xi \mathbb{1} - A), \qquad p(A) = \sum_{j=0}^{N} c_j A^j = 0. \tag{158}$$

Coefficients $c_j$ are provided by traces of powers of $A$, $a_j \equiv \text{Tr} A^j$,

$$c_{N-j} = \frac{(-1)^j}{j!} B_j \Big( 0! a_1, -1! a_2, 2! a_3, \ldots, (-1)^j (j-1)! a_j \Big), \tag{159}$$

with $B_j$ denoting the exponential Bell polynomials. The key observation here is that a matrix of dimension $N$ possesses only $N$ independent scalar invariants (for instance $a_j$ up to $j = N$). Specifically, we define the Cayley–Hamilton polynomial corresponding to the signature $\Sigma^{(k,N)}$ as

$$p_{k,N}(\xi) = \text{Det}(\xi \mathbb{1} - S_0). \tag{160}$$

We can accordingly proceed by solving the following truncated system of partial differential equations

$$\sum_{m=1}^{\lfloor N/2 \rfloor} 2m \frac{\partial \mathscr{H}_\tau^{(k,N)}}{\partial s_{2m}} s_{2m+n-1} = \frac{i}{\tau} \sum_{j=1}^{\infty} \frac{(-i\tau)^j}{j} s_{n-j}, \quad n \in \mathbb{Z}. \tag{161}$$

We shall not attempt to find its general solution here, but instead rather consider the few simplest instances for small matrix dimensions $N$.

**Case $N = 2$.** In the case of $2 \times 2$ matrices we have $s_0 = \text{Tr} \mathbb{1} = 2$ and there is a single nontrivial signature $s_1 = 0$, implying that all positive odd $s_k$ vanish as well. The system (161) evaluated at $m = -1$ simplifies to

$$4 \frac{\partial \mathscr{H}_\tau^{(1,2)}}{\partial s_2} = \frac{i}{\tau} \sum_{j=1}^{\infty} \frac{(-i\tau)^j}{j} s_{-j-1}. \tag{162}$$

Invoking the Cayley-Hamilton theorem for $N = 2$,

$$p(A) = A^2 - \text{Tr}(A) A - \frac{1}{2} \Big( \text{Tr}(A^2) - \text{Tr}(A)^2 \Big) \mathbb{1} = 0, \tag{163}$$

we arrive at a simple recursion relation for $s_k$

$$s_{-k} = \frac{2 s_{2-k}}{s_2}, \tag{164}$$

with the solution

$$s_{-2k+1} = 0, \qquad s_{-2k} = 2 (s_2/2)^{-k}. \tag{165}$$

This allows us to perform the resummation of the right-hand side of Eq. (162),

$$\frac{\partial \mathscr{H}_\tau^{(1,2)}}{\partial s_2} = \frac{1}{2\tau} \frac{1}{\tilde{s}} \arctan\left(\frac{\tau}{\tilde{s}}\right), \qquad \tilde{s} = \sqrt{s_2/2}, \tag{166}$$

which can be readily integrated

$$\mathscr{H}_\tau^{(1,2)} = \log\left(\tilde{s}^2 + \tau^2\right) + \frac{2\tilde{s}}{\tau} \arctan\left(\frac{\tau}{\tilde{s}}\right). \tag{167}$$

This is in agreement with the expression found previously in [24]. In the $\tau \to 0$ limit, this yields (modulo a constant term) the Hamiltonian of the isotropic Landau-Lifshitz ferromagnet

$$\lim_{\tau \to 0} \mathscr{H}_\tau^{(1,2)} \simeq \log \text{Tr}\left((M_1 + M_2)^2\right). \tag{168}$$

**Case $N = 3$.** Now $s_0 = 3$, and there are two possible signatures to consider: $s_1 = 2$ ($k = 1$) and $s_1 = -2$ ($k = 2$). The system of equations (161) reduces to

$$6\frac{\partial \mathcal{H}_\tau^{(k,3)}}{\partial s_2} = \frac{i}{\tau}\sum_{j=1}^{\infty}\frac{(-i\tau)^j}{j}s_{-j-1}. \tag{169}$$

With the help of the Cayley-Hamilton polynomial for $3 \times 3$ matrices,

$$A^3 - \text{Tr}(A)A^2 - \frac{1}{2}\big(\text{Tr}(A^2) - \text{Tr}(A)^2\big)A - \frac{1}{6}\big((\text{Tr}A)^3 - 3\text{Tr}(A^2)\text{Tr}A + 2\text{Tr}(A^3)\big)\mathbb{1} = 0, \tag{170}$$

we find the following recursion relation

$$s_{3-m} \mp 2s_{2-m} - \frac{1}{2}(s_2 - 4)s_{1-m} \pm (s_2 - 4)s_{-m} = 0, \tag{171}$$

with the solution

$$s_{-m} = 2^{m/2}\big((\pm 1)^m 2^{-3m/2} + (1 + (-1)^m)(s_2 - 4)^{-m/2}\big). \tag{172}$$

The resummation of the infinite sum in the right-hand side of Eq. (169) now yields

$$6\frac{\partial \mathcal{H}_\tau^{(k,3)}}{\partial s_2} = \frac{1}{2\tau}\Big[\frac{1}{\tilde{s}}\arctan\Big(\frac{\tau}{\tilde{s}}\Big) \mp i\log\big(1 \pm i\tau/2\big)\Big], \qquad \tilde{s} = \sqrt{(s_2 - 4)/2}, \tag{173}$$

which can be readily integrated

$$\mathcal{H}_\tau^{(k,3)} = \frac{2}{3}\Big[\log\big(\tilde{s}^2 + \tau^2\big) + \frac{2\tilde{s}}{\tau}\arctan\Big(\frac{\tau}{\tilde{s}}\Big)\Big] \mp i\frac{\tilde{s}^2 + 2}{6\tau}\log\big(1 \pm i\tau/2\big), \quad k = 1, 2. \tag{174}$$

The first two terms in this expression exactly match the form of the $N = 2$ case above, apart from a different multiplicative factor in front. Somewhat unexpectedly, the symplectic generator involves an imaginary term which only reduces to a real quantity in the $\tau \to 0$ limit.

**Case $N = 4$.** We conclude our analysis by considering also the $N = 4$ case, which represents the first instance where the generator involves two functionally independent matrix invariants $s_2$ and $s_4$. Let us specialize to the traceless case with $s_{2j-1} = 0$, i.e. signature $\Sigma^{(2,4)}$. Eq. (161) thus provides us with two independent equations

$$8\frac{\partial \mathcal{H}_\tau^{(2,4)}}{\partial s_2} + 4\frac{\partial \mathcal{H}_\tau^{(2,4)}}{\partial s_4}s_2 = \frac{i}{\tau}\sum_{j=1}^{\infty}\frac{(i\tau)^j}{j}s_{-j-1}, \tag{175}$$

$$2\frac{\partial \mathcal{H}_\tau^{(2,4)}}{\partial s_2}s_2 + 4\frac{\partial \mathcal{H}_\tau^{(2,4)}}{\partial s_4}s_4 = \frac{i}{\tau}\sum_{j=1}^{\infty}\frac{(i\tau)^j}{j}s_{-j+1}. \tag{176}$$

Invoking once again the Cayley–Hamilton theorem, this time using that the odd trace invariants are all zero, we arrive at the recursion of the form

$$s_{k+4} - \frac{1}{2}s_2 s_{2+k} + \frac{1}{24}(3s_2^2 - 6s_4)s_k = 0, \quad s_0 = 8, \quad s_1 = 0. \tag{177}$$

The solution can still be found in closed form,

$$s_{2m} = 2\big((s_+)^{2m} + (s_-)^{2m}\big), \qquad s_\pm = \frac{1}{2}\sqrt{s_2 \pm \sqrt{4s_4 - s_2^2}}, \tag{178}$$

whereas $s_{2m+1} = 0$. Performing the resummation, we find a system of two coupled PDEs

$$2\frac{\partial \mathscr{H}_\tau^{(2,4)}}{\partial s_2} + \frac{\partial \mathscr{H}_\tau^{(2,4)}}{\partial s_4} s_2 = \frac{1}{\tau}\Big(\frac{1}{s_-}\arctan(\tau/s_-) + \frac{1}{s_+}\arctan(\tau/s_+)\Big), \qquad (179)$$

$$\frac{\partial \mathscr{H}_\tau^{(2,4)}}{\partial s_2} s_2 + 2\frac{\partial \mathscr{H}_\tau^{(2,4)}}{\partial s_4} s_4 = \frac{1}{\tau}\Big(s_-\arctan(\tau/s_-) + s_+\arctan(\tau/s_+)\Big). \qquad (180)$$

We can now solve for $\partial \mathscr{H}_\tau^{(2,4)}/\partial s_4$, and after subsequently integrating the result we obtain

$$\int ds_4 \frac{\partial \mathscr{H}_\tau^{(2,4)}}{\partial s_4} = \mathscr{H}_\tau^{(2,4)} + \gamma(s_2), \qquad (181)$$

with

$$\mathscr{H}_\tau^{(2,4)} = \sum_{\alpha=\pm}\left[\log(s_\alpha^2 + \tau^2) + \frac{2s_\alpha}{\tau}\arctan\Big(\frac{\tau}{s_\alpha}\Big)\right], \qquad (182)$$

uniquely up to an additive constant. By computing the partial derivative $\partial \mathscr{H}_\tau^{(2,4)}/\partial s_2$ and comparing it to the solution to Eqs. (180), we conclude that the undetermined function $\gamma$ is in fact a constant, namely independent of $s_2$. Expression (182) is therefore the final form of the symplectic generator $\mathscr{H}_\tau^{(2,4)}$. The result is indeed in agreement with expression (167) found in the $N = 2$ case, the only difference being that here the additional (double) root of the Cayley–Hamilton polynomial enters and that invariants now take a different functional form.

**Conjecture.** Despite the fact that we have not managed to find a general procedure for reducing and solving the infinite system of PDEs (157), we can nonetheless conjecture, based on the above considerations, the general form of the symplectic generators for the case of traceless ($s_1 = 0$) even-dimensional matrices for general even $N$, i.e. for signature matrices $\Sigma^{(N/2,N)}$. Since all odd invariants $s_{2m+1}$ vanish, the corresponding Cayley–Hamilton polynomials involves only double roots $\{\tilde{s}_j\}_{j=1}^{N/2}$, $p_{N/2,N}(\tilde{s}_j) = p'_{N/2,N}(\tilde{s}_j) = 0$. We conjecture the solutions to Eqs. (161) in this case take the form (uniquely, modulo additive constants)

$$\mathscr{H}_\tau^{(N/2,N)} = \sum_{j=1}^{N/2}\left(\log\big(\tau^2 + \tilde{s}_j^2\big) + \frac{2\tilde{s}_j}{\tau}\arctan\Big(\frac{\tau}{\tilde{s}_j}\Big)\right). \qquad (183)$$

## C   Lax representations

In this section we collect all the Lax (zero-curvature) representations for (i) discrete space and time, (ii) continuous time and discrete space, and (iii) continuous space-time models.

First we shortly recall the space-time discrete zero-curvature representation for the light-cone Lax matrix $L(\lambda; M) = \lambda\mathbb{1} + iM$ with $M \in \mathcal{M}_1$ satisfying the constraint $M^2 = \mathbb{1}$,

$$F\,L(\lambda; M_2)L(\mu; M_1) = L(\mu; M_2')L(\lambda; M_1')\,F, \qquad (184)$$

which admits a unique solution $\Phi_\tau : (M_1, M_2) \mapsto (M_1', M_2')$,

$$M_1' = \mathrm{Ad}_{FS_\tau}(M_2), \qquad M_2' = \mathrm{Ad}_{FS_\tau}(M_1), \qquad S_\tau \equiv M_1 + M_2 + i\tau\mathbb{1}, \qquad (185)$$

where $F$ is a constant invertible complex matrix and $\mathrm{Ad}_A(B) = ABA^{-1}$.

**Semi-discrete limit.** In the continuous time limit, $\tau \to 0$, the symplectic maps reduces to the following equation of motion

$$\frac{\mathrm{d}M_\ell}{\mathrm{d}t} = -\mathrm{i}\big[M_\ell, (M_{\ell-1} + M_\ell)^{-1} + (M_\ell + M_{\ell+1})^{-1} - B\big]. \tag{186}$$

The obtained equation of motion is generated by the Hamiltonian $H_{\mathrm{lattice}}$, reading

$$\frac{\mathrm{d}M_\ell}{\mathrm{d}t} = \{M_\ell, H_{\mathrm{lattice}}\}, \qquad H_{\mathrm{lattice}} \simeq \sum_{\ell=1}^{L} \Big(\mathrm{Tr}\big(M_\ell B\big) - \mathrm{Tr}\log(M_\ell + M_{\ell+1})\Big), \tag{187}$$

where the Poisson bracket is defined as

$$\big\{M_\ell \overset{\otimes}{,} M_{\ell'}\big\} = -\frac{\mathrm{i}}{2}\Big[\Pi, M_\ell \otimes \mathbb{1}_N - \mathbb{1}_N \otimes M_\ell\Big]\delta_{\ell,\ell'}. \tag{188}$$

Equation (186) plays the role of a compatibility condition for an auxiliary linear problem in the form of a semi-discrete zero-curvature condition, see (98) in the main text.

For the subsequent derivation we omit the twist dependence and temporarily put $F = \mathbb{1}$ (equiv. $B = 0$); the latter can be easily incorporated back at the very end of computation. The first step is to promote the Poission structure (188) to the level of Lax matrices which yields the quadratic Sklyanin bracket

$$\big\{L_\ell(\lambda) \overset{\otimes}{,} L_{\ell'}(\lambda')\big\} = \big[r(\lambda, \lambda'), L_\ell(\lambda) \otimes L_{\ell'}(\mu)\big]\delta_{\ell,\ell'}, \qquad r(\lambda, \lambda') = \frac{\Pi}{\lambda' - \lambda}, \tag{189}$$

where we have used a short-hand notation $L_\ell(\lambda) \equiv L(\lambda; M_\ell)$. The time evolution of the Lax matrix reads by definition

$$\frac{\mathrm{d}}{\mathrm{d}t}L_\ell(\lambda) = \{L_\ell(\lambda), H_{\mathrm{lattice}}\} = -\{L_\ell(\lambda), \mathrm{Tr}\log(M_{\ell-1} + M_\ell) + \mathrm{Tr}\log(M_\ell + M_{\ell+1})\}. \tag{190}$$

Writing $L_\ell^\pm = L_\ell(\pm\mathrm{i}) = \mathrm{i}(\pm\mathbb{1} + M_\ell)$ and introducing double-site matrices[15]

$$A_{\ell-1,\ell} = 2\mathrm{i}(M_{\ell-1} + M_\ell) = L_{\ell-1}^+ + L_{\ell-1}^- + L_\ell^+ + L_\ell^-, \tag{191}$$

the above expression can be put in the form (whilst dropping a constant term)

$$\frac{\mathrm{d}}{\mathrm{d}t}L_\ell(\lambda) = -\{L_\ell(\lambda), \mathrm{Tr}\log(A_{\ell-1,\ell}) + \mathrm{Tr}\log(A_{\ell,\ell+1})\}. \tag{192}$$

By direct computation we then find

$$\frac{\mathrm{d}}{\mathrm{d}t}L_\ell(\lambda) = -\mathrm{Tr}_2\Big(\big(\mathbb{1} \otimes (A_{\ell-1,\ell}^{-1} + A_{\ell,\ell+1}^{-1})\big)\{L_\ell(\lambda) \otimes (L_\ell^+ + L_\ell^-)\}\Big). \tag{193}$$

Making use of the Sklyanin bracket and the identity $\mathrm{Tr}_2\big([\Pi, A \otimes B](I \otimes C)\big) = BCA - ACB$, the above expression can be brought into the form

$$\frac{\mathrm{d}}{\mathrm{d}t}L_\ell(\lambda) = -\frac{1}{\mathrm{i} - \lambda}L_\ell^+\big(A_{\ell-1,\ell}^{-1} + A_{\ell,\ell+1}^{-1}\big)L_\ell(\lambda) - L_\ell(\lambda)\big(A_{\ell-1,\ell}^{-1} + A_{\ell,\ell+1}^{-1}\big)L_\ell^+$$
$$+ \frac{1}{\mathrm{i} + \lambda}L_\ell^-\big(A_{\ell-1,\ell}^{-1} + A_{\ell,\ell+1}^{-1}\big)L_\ell(\lambda) - L_\ell(\lambda)\big(A_{\ell-1,\ell}^{-1} + A_{\ell,\ell+1}^{-1}\big)L_\ell^-. \tag{194}$$

---

[15]We note that this representation of operator $A$ is not unique, but has been adopted for symmetry reasons. Notice moreover that $A$ is, apart from normalization, equal to $S_{\tau=0}$.

As a consequence of $M^2 = \mathbb{1}$, the following swap operations hold

$$(M_1 + M_2)M_{1,2} = M_{2,1}(M_1 + M_2), \qquad M_{1,2}(M_1 + M_2)^{-1} = (M_1 + M_2)^{-1}M_{2,1}. \tag{195}$$

Exploiting the above identities we have

$$A_{\ell-1,\ell}^{-1}L_\ell(\lambda) = L_{\ell-1}(\lambda)A_{\ell-1,\ell}^{-1}, \qquad L_\ell(\lambda)A_{\ell-1,\ell}^{-1} = A_{\ell-1,\ell}^{-1}L_{\ell-1}(\lambda), \tag{196}$$

which readily yields a (non-standard) semi-discrete zero-curvature representation

$$\frac{\mathrm{d}}{\mathrm{d}t}L_\ell(\lambda) = V_{\ell+1}^{\mathrm{L}}(\lambda)L_\ell(\lambda) - L_\ell(\lambda)V_\ell^{\mathrm{L}}(\lambda) + L_\ell(\lambda)V_{\ell+1}^{\mathrm{R}}(\lambda) - V_\ell^{\mathrm{R}}(\lambda)L_\ell(\lambda), \tag{197}$$

with 'left' and 'right' temporal propagators

$$V_\ell^{\mathrm{L}}(\lambda) = A_{\ell-1,\ell}^{-1}\left(\frac{L_\ell^-}{\mathrm{i}+\lambda} - \frac{L_\ell^+}{\mathrm{i}-\lambda}\right) = \frac{2\lambda}{\lambda^2+1}A_{\ell-1,\ell}^{-1}L_\ell(-\lambda^{-1}) \tag{198}$$

$$V_\ell^{\mathrm{R}}(\lambda) = A_{\ell-1,\ell}^{-1}\left(\frac{L_{\ell-1}^+}{\mathrm{i}-\lambda} - \frac{L_{\ell-1}^-}{\mathrm{i}+\lambda}\right) = -\frac{2\lambda}{\lambda^2+1}A_{\ell-1,\ell}^{-1}L_{\ell-1}(-\lambda^{-1}). \tag{199}$$

The last step is to transform Eq. (197) into the standard form

$$\frac{\mathrm{d}}{\mathrm{d}t}L_\ell(\lambda) = V_{\ell+1}(\lambda)L_\ell(\lambda) - L_\ell(\lambda)V_\ell(\lambda), \tag{200}$$

from where we can determine the temporal component of the Lax pair $V_{\ell,\ell+1}(\lambda)$. This can be accomplished with aid of the following 'inversion identities' for the Lax matrices,

$$L_\ell(\lambda)L_{\ell'}(-\lambda^{-1}) = L_\ell(\lambda)L_{\ell'}(\lambda) - (\lambda+\lambda^{-1})L_\ell(\lambda), \tag{201}$$
$$L_\ell(-\lambda^{-1})L_{\ell'}(\lambda) = L_\ell(\lambda)L_{\ell'}(\lambda) - (\lambda+\lambda^{-1})L_{\ell'}(\lambda), \tag{202}$$

which combined imply the 'exchange relation',

$$L_\ell(\lambda)L_{\ell'}(-\lambda^{-1}) + L_\ell(-\lambda^{-1})L_{\ell'}(\lambda) = 2L_\ell(\lambda)L_{\ell'}(\lambda) - (\lambda+\lambda^{-1})\bigl(L_\ell(\lambda) + L_{\ell'}(\lambda)\bigr). \tag{203}$$

Using these, the right propagators appearing in $L_\ell(\lambda)V_{\ell+1}^{\mathrm{R}}(\lambda) - V_\ell^{\mathrm{R}}(\lambda)L_\ell(\lambda)$ in Eq. (197) can be brought to the left in the following manner:

$$\begin{aligned}
&L_\ell(\lambda)V_{\ell+1}^{\mathrm{R}}(\lambda) - V_\ell^{\mathrm{R}}(\lambda)L_\ell(\lambda) \\
&= -\frac{2\lambda}{1+\lambda^2}\Bigl(L_\ell(\lambda)A_{\ell,\ell+1}^{-1}L_\ell(-\lambda^{-1}) - A_{\ell-1,\ell}^{-1}L_{\ell-1}(-\lambda^{-1})L_\ell(\lambda)\Bigr) \\
&= -\frac{2\lambda}{1+\lambda^2}\Bigl(A_{\ell,\ell+1}^{-1}L_{\ell+1}(\lambda)L_\ell(-\lambda^{-1}) - A_{\ell-1,\ell}^{-1}L_{\ell-1}(-\lambda^{-1})L_\ell(\lambda)\Bigr) \\
&= -\frac{2\lambda}{1+\lambda^2}\Bigl(A_{\ell,\ell+1}^{-1}\bigl(2L_{\ell+1}(\lambda)L_\ell(\lambda) - L_{\ell+1}(-\lambda^{-1})L_\ell(\lambda) - (\lambda+\lambda^{-1})(L_{\ell+1}(\lambda) + L_\ell(\lambda))\bigr) \\
&\quad - A_{\ell-1,\ell}^{-1}\bigl(2L_{\ell-1}(\lambda)L_\ell(\lambda) - L_{\ell-1}(\lambda)L_\ell(-\lambda^{-1}) - (\lambda+\lambda^{-1})(L_{\ell-1}(\lambda) + L_\ell(\lambda))\bigr)\Bigr) \\
&= \frac{2\lambda}{1+\lambda^2}\Bigl(A_{\ell,\ell+1}^{-1}\bigl(L_{\ell+1}(-1/\lambda) - 2L_{\ell+1}(\lambda)\bigr)L_\ell(\lambda) - L_\ell(\lambda)A_{\ell-1,\ell}^{-1}\bigl(L_\ell(-1/\lambda) - 2L_\ell(\lambda)\bigr) \\
&\quad + 2(A_{\ell,\ell+1}^{-1}\bigl(L_{\ell+1}(\lambda) + L_\ell(\lambda)\bigr) - A_{\ell-1,\ell}^{-1}\bigl(L_{\ell-1}(\lambda) + L_\ell(\lambda)\bigr)\Bigr).
\end{aligned} \tag{204}$$

The first two terms in the obtained expression are already in the desired form. The last two terms can be further simplified with the use of $\mathbb{1} = -(\lambda^2+1)^{-1}L_\ell(-\lambda)L_\ell(\lambda)$, yielding

$$L_\ell(\lambda)V_{\ell+1}^{\mathrm{R}}(\lambda) - V_\ell^{\mathrm{R}}(\lambda)L_\ell(\lambda) = \widetilde{V}_{\ell+1}(\lambda)L_\ell(\lambda) - L_\ell(\lambda)\widetilde{V}_\ell, \tag{205}$$

with

$$\widetilde{V}_\ell(\lambda) = \frac{2\lambda}{1+\lambda^2}\left(A^{-1}_{\ell-1,\ell}\left(L_\ell(-\lambda^{-1}) - 2L_\ell(\lambda) - 2L_{\ell-1}(-\lambda)\right)\right). \tag{206}$$

By substituting this result back into the 'non-canonical' zero-curvature condition (197), we eventually restore the standard form (200), with the temporal component $V_\ell(\lambda)$ reading

$$V_\ell(\lambda) = \frac{4\lambda}{1+\lambda^2}\left(A^{-1}_{\ell-1,\ell}(L_\ell(-\lambda^{-1}) - L_\ell(\lambda) - L_{\ell-1}(-\lambda))\right) + \mathrm{i}B$$

$$= -\frac{4\lambda}{\lambda^2+1}A^{-1}_{\ell-1,\ell}L_{\ell-1}(\lambda^{-1}) + \mathrm{i}B \tag{207}$$

$$= \frac{-2\lambda}{\lambda^2+1}\left(L_\ell(0) + L_{\ell-1}(0)\right)^{-1}L_{\ell-1}(\lambda^{-1}) + \mathrm{i}B, \tag{208}$$

where we have simultaneously reinstated the magnetic field.

**Zero-curvature formulation in continuous space-time.** Having derived the semi-discrete version of the zero-curvature representation, we are in a position to take the continuum theory limit and infer also the zero-curvature representation of the PDE

$$M_t = \{M(x,t), H_c\} = \frac{1}{2\mathrm{i}}\left[M, M_{xx}\right] + \mathrm{i}[B, M]. \tag{209}$$

First we recall the continuum counterpart of the lattice Hamiltonian (187),

$$H_c = \int \mathrm{d}x\left[\frac{1}{4}\mathrm{Tr}(M_x^2) + \mathrm{Tr}(MB)\right], \tag{210}$$

which (using the Poisson bracket (97)) generates Eq. (209).

This is achieved in the usual manner by expanding the semi-discrete zero-curvature condition (200) to lowest non-trivial order in lattice spacing $\Delta$ and regarding lattice variables $M_\ell(t)$ as smoothly varying field configurations $M(x,t)$, namely $M_{\ell+1} = M + \Delta M_x + \frac{\Delta^2}{2}M_{xx} + \mathcal{O}(\Delta^3)$. Expanding Eq. (200) to the quadratic order $\mathcal{O}(\Delta^2)$, we find

$$\mathrm{i}M_t = \frac{\Delta^2}{4}(MM_{xx} - M_{xx}M) - [B, M] = \frac{\Delta^2}{2}(MM_{xx} - M_x^2) - [B, M], \tag{211}$$

where we have used $MM_x = -M_x M$ and $MM_{xx} + M_{xx}M = -2M_x^2$. Dividing subsequently by $\lambda$ and simultaneously rescaling time and magnetic field as $t \to (2/\Delta^2)t$, $B \to (\Delta^2/2)B$, we arrive at the following Lax pair

$$\mathscr{U}(\lambda; x, t) = \frac{\mathrm{i}}{\lambda}M, \qquad \mathscr{V}(\lambda; x, t) = \frac{2\mathrm{i}}{\lambda^2}M - \frac{1}{\lambda}M_x M + \mathrm{i}B, \tag{212}$$

satisfying the continuous version of the zero-curvature condition

$$\partial_t\mathscr{U} - \partial_x\mathscr{V} + [\mathscr{U}, \mathscr{V}] = 0. \tag{213}$$

# D  Local conservation laws

One of the central implications of the Lax (zero-curvature) property is isospectrality, see Section 2.4.1. As a direct corollary, the model possesses $\mathcal{O}(L)$ functionally independent conserved phase-space functions in involution. In integrable systems with local interactions one can typically extract local conservation laws, i.e. conserved quantities that can be represented as a spatially homogeneous sum of densities with a compact support. A common procedure to infer

the local charge is to expand the logarithm of the transfer map as a power series in $\lambda$ around a distinguished point $\lambda_0$. In the context of integrable matrix models introduced in Section 2, natural candidates for such expansion points are

$$\lambda_0 \in \big\{ \pm \mathrm{i}, \pm \mathrm{i} - \tau \big\}. \tag{214}$$

At these values one of the Lax matrices $L(\lambda)$ or $L(\lambda + \tau)$ in the staggered monodromy $\mathbb{M}(\lambda, \mu; \{M_\ell\})$ degenerates into a projector,

$$L(\mp \mathrm{i}) = \mp 2\mathrm{i} \, P^{(\pm)}. \tag{215}$$

Here $P^{(+)} \equiv P$ denotes a rank-$k$ projector and $P^{(-)} = \mathbb{1}_N - P^{(+)}$ is its orthogonal complement of rank $N - k$. In what follows we assume $k \leq N/2$, in which case the Lax matrices degenerate into $P^{(+)}$.

Employing the projector realization (18) with Eq. (27) and evaluating the transfer matrices at the appropriate projection points, we obtain

$$T_\tau^{\mathrm{odd}}(-\mathrm{i}) = (-2\mathrm{i})^{L/2} \operatorname{Tr} \prod_{\ell=1}^{L/2} P_0 \, g_{2\ell+1}^{-1} L_{2\ell}(-\mathrm{i} + \tau) g_{2\ell-1}, \tag{216}$$

$$T_\tau^{\mathrm{even}}(-\mathrm{i} - \tau) = (-2\mathrm{i})^{L/2} \operatorname{Tr} \prod_{\ell=1}^{L/2} P_0 \, g_{2\ell+2}^{-1} L_{2\ell+1}(-\mathrm{i} - \tau) g_{2\ell}, \tag{217}$$

where $P_0$ and $g_\ell$ are defined in Eq. (18).

Local conserved quantities are typically produced by taking the logarithm of Poisson-commuting transfer matrices. In general, it is not manifest from the above expressions that their logarithms will generate strictly local object. The exception are only the $k = 1$ cases where the transfer matrices evaluated at $\lambda_0 = -\mathrm{i}$ yield completely factorizable scalar expressions which are considered below.

**Local conservations laws in projective models.** We subsequently assume $k = 1$, corresponding to integrable matrix models with complex projective spaces $\mathbb{CP}^{N-1}$ as target spaces. The associated Lax matrices at two distinguished points $\lambda_0 = \pm \mathrm{i}$ degenerate into rank-1 projectors

$$L(\mp \mathrm{i})_\ell = \mp 2\mathrm{i} \, P_\ell^{(\pm)}, \qquad P_\ell^{(\pm)} = |\Psi_\ell^{(\pm)}\rangle \langle \Psi_\ell^{(\pm)}|. \tag{218}$$

Let us remind that spatial subscripts, such as in $\Psi_\ell^\pm$, designate that the quantity depends on the local matrix variable $M_\ell$. As a consequence, the transfer maps completely factorize into sequences of scalars (matrix elements)

$$T_\tau^{\mathrm{odd}}(-\mathrm{i}) = (-2\mathrm{i})^{L/2} \prod_{\ell=1}^{L/2} \langle \Psi_{2\ell+1}^{(+)} | \, L_{2\ell}(-\mathrm{i} + \tau) \, | \Psi_{2\ell-1}^{(+)} \rangle, \tag{219}$$

$$T_\tau^{\mathrm{even}}(-\mathrm{i} - \tau) = (-2\mathrm{i})^{L/2} \prod_{\ell=1}^{L/2} \langle \Psi_{2\ell+2}^{(+)} | \, L_{2\ell+1}(-\mathrm{i} - \tau) \, | \Psi_{2\ell}^{(+)} \rangle. \tag{220}$$

Here the spatial index should be understood modulo $L$, whereas the nomenclature 'even' and 'odd' here refers to the sub-lattices at which the degeneracy occurs. By exploiting the conjugation property of the Lax matrix,

$$[L(\lambda; M)]^\dagger = -L\big( -\bar{\lambda}; M \big), \tag{221}$$

we readily obtain the complex-conjugate counterparts of Eqs. (219) and (220),

$$\overline{T_\tau^{\mathrm{odd}}(-\mathrm{i})} = (2\mathrm{i})^{L/2} \prod_{\ell=1}^{L/2} \langle \Psi_{2\ell-1}^{(+)} | L_{2\ell}(-\mathrm{i}-\tau) | \Psi_{2\ell+1}^{(+)} \rangle, \tag{222}$$

$$\overline{T_\tau^{\mathrm{even}}(-\mathrm{i}-\tau)} = (2\mathrm{i})^{L/2} \prod_{\ell=1}^{L/2} \langle \Psi_{2\ell}^{(+)} | L_{2\ell+1}(-\mathrm{i}+\tau) | \Psi_{2\ell+2}^{(+)} \rangle. \tag{223}$$

As an immediate corollary of this construction, the logarithms of the square moduli of the transfer matrices yield conserved quantities which are manifestly *local* and real,

$$Q_\tau^{\mathrm{odd}(1)} = \log \left| T_\tau^{\mathrm{odd}}(-\mathrm{i}) \right|^2 = \sum_{\ell=1}^{L/2} q_\ell^{\mathrm{odd}(1)}(\tau), \tag{224}$$

$$Q_\tau^{\mathrm{even}(1)} = \log \left| T_\tau^{\mathrm{even}}(-\mathrm{i}-\tau) \right|^2 = \sum_{\ell=1}^{L/2} q_\ell^{\mathrm{even}(1)}(\tau), \tag{225}$$

with densities

$$q_\ell^{\mathrm{odd}(1)}(\tau) = \log \mathrm{Tr}\Big( L_{2\ell+1}(-\mathrm{i}) L_{2\ell}(-\mathrm{i}+\tau) L_{2\ell-1}(-\mathrm{i}) L_{2\ell}(\mathrm{i}-\tau) \Big), \tag{226}$$

$$q_\ell^{\mathrm{even}(1)}(\tau) = \log \mathrm{Tr}\Big( L_{2\ell+2}(-\mathrm{i}) L_{2\ell+1}(-\mathrm{i}-\tau) L_{2\ell}(-\mathrm{i}) L_{2\ell+1}(-\mathrm{i}+\tau) \Big). \tag{227}$$

An infinite tower of higher local conservation laws, labelled by integer $n \in \mathbb{N}$, can then be produced in an iterative fashion by means of logarithmic differentiation,

$$Q_\tau^{\mathrm{odd}(n)} = \partial_\lambda^{n-1} \log |T_\tau(\lambda)|^2 \Big|_{\lambda=-\mathrm{i}}, \qquad Q_\tau^{\mathrm{even}(n)} = \partial_\lambda^{n-1} \log |T_\tau(\lambda)|^2 \Big|_{\lambda=-\mathrm{i}-\tau}. \tag{228}$$

In this way, we obtain two inequivalent sequences of local charges with densities supported on $2n+1$ adjacent lattice sites.[16] It is worthwhile stressing here that there is no connection between the lowest local conservation law in the hierarchy (whose density is supported on three adjacent lattice sites) and the symplectic two-body propagator.

**Hamiltonian limit.** In the continuous time limit $\tau \to 0$ the degeneracy points $\lambda_0 = \{-\mathrm{i}, -\mathrm{i}-\tau\}$ collide with one another and the two towers of local conservation laws merge together. This means in effect that an infinite family of local conservation laws can now be generated from the logarithm of the homogeneous transfer map and $\lambda$-derivatives thereof. In particular, for the lowest-degree conservation laws we find,

$$\lim_{\tau \to 0} Q_\tau^{\mathrm{even}(1)} = \lim_{\tau \to 0} Q_\tau^{\mathrm{odd}(1)} = L \log(4) + \sum_{\ell=1}^{L} \log \mathrm{Tr}\Big( P_{\ell+1}^{(+)} P_\ell^{(+)} \Big), \tag{229}$$

yielding the Hamiltonian density in the form of the logarithmic overlap,

$$h_\ell^{(1)} = \log \mathrm{Tr}\Big( P_{\ell+1}^{(+)} P_\ell^{(+)} \Big) = \log \left| \langle \Psi_{\ell+1}^{(+)} | \Psi_\ell^{(+)} \rangle \right|^2. \tag{230}$$

Below we give an explicit parametrization using local affine coordinates

$$\mathbf{z}_\ell = (z_{1;\ell}, z_{2;\ell}, \ldots, z_{N-1;\ell})^{\mathrm{T}}, \tag{231}$$

---

[16]The outlined construction is only meaningful as long as the support of the densities does not exceed the length of the system, i.e. before 'wrapping effects' take place.

in terms of which

$$|\Psi_\ell^{(+)}\rangle = (1 + \langle \mathbf{z}_\ell | \mathbf{z}_\ell \rangle)^{-1/2}(1, z_{\ell;1}, \ldots, z_{\ell;N-1})^{\mathrm{T}}, \tag{232}$$

with overlap coefficient $\langle \mathbf{z}_\ell | \mathbf{z}_{\ell'} \rangle \equiv \bar{\mathbf{z}}_\ell \cdot \mathbf{z}_{\ell'} = \sum_{j=1}^{N-1} \bar{z}_{j;\ell} z_{j;\ell'}$. The rank-1 projector $P_\ell^{(+)}$ then reads

$$P_\ell^{(+)}(\mathbf{z}) = (1 + \langle \mathbf{z}_\ell | \mathbf{z}_\ell \rangle)^{-1} \begin{pmatrix} 1 & \bar{\mathbf{z}}_\ell^{\mathrm{T}} \\ \mathbf{z}_\ell & \mathbf{z}_\ell \cdot \bar{\mathbf{z}}_\ell^{\mathrm{T}} \end{pmatrix}, \tag{233}$$

and the leading density given by Eq. (230) takes the form

$$h_\ell^{(1)} = \log \frac{(1 + \langle \mathbf{z}_{\ell+1} | \mathbf{z}_\ell \rangle)(1 + \langle \mathbf{z}_\ell | \mathbf{z}_{\ell+1} \rangle)}{(1 + \langle \mathbf{z}_\ell | \mathbf{z}_\ell \rangle)(1 + \langle \mathbf{z}_{\ell+1} | \mathbf{z}_{\ell+1} \rangle)}. \tag{234}$$

**Example.** For illustration we consider the $\mathbb{CP}^1 \cong S^2$ case. The affine complex variable which parametrizes the corresponding rank-1 projectors $P_\ell(z)$ is the local stereographic coordinate $z_\ell = \tan(\theta_\ell/2)\exp(\mathrm{i}\phi_\ell)$. We thus have

$$P_\ell(z) = \begin{pmatrix} 1 & -\bar{z}_\ell \\ z_\ell & 1 \end{pmatrix} \begin{pmatrix} 1 & 0 \\ 0 & 0 \end{pmatrix} \begin{pmatrix} 1 & -\bar{z}_\ell \\ z_\ell & 1 \end{pmatrix}^{-1} = \frac{1}{1+|z_\ell|^2} \begin{pmatrix} 1 & \bar{z}_\ell \\ z_\ell & |z_\ell|^2 \end{pmatrix}. \tag{235}$$

The outcome is the Hamiltonian density of the isotropic lattice Landau–Lifshitz model [85,108]

$$\mathbb{CP}^1: \qquad h_\ell^{(1)} = \log \frac{1 + \bar{z}_{\ell+1} z_\ell + z_{\ell+1} \bar{z}_\ell + |z_\ell|^2 |z_{\ell+1}|^2}{(1+|z_\ell|^2)(1+|z_{\ell+1}|^2)} = \log\left[\frac{1}{2}(1 + \mathbf{S}_\ell \cdot \mathbf{S}_{\ell+1})\right], \tag{236}$$

where in the second line we have used the spin-field realization

$$\mathbf{S}_\ell = (\sin(\theta_\ell)\cos(\phi_\ell), \sin(\theta_\ell)\sin(\phi_\ell), \cos(\theta_\ell))^{\mathrm{T}}. \tag{237}$$

Reintroducing the lattice spacing $\Delta$ and performing the long-wavelength expansion,

$$\mathbf{S}_\ell(t) \to \mathbf{S}(x = \ell\Delta, t), \qquad \mathbf{S}_{\ell+1}(t) \to \mathbf{S}(x = \ell\Delta, t) + \Delta\, \mathbf{S}_x(x, t) + \frac{\Delta^2}{2}\mathbf{S}_{xx}(x, t) + \mathcal{O}(\Delta^3), \tag{238}$$

we find, at the leading order $\mathcal{O}(\Delta^2)$ and with rescaling time $(\Delta^2/2)t \to J t$, the Hamiltonian of the Landau–Lifshitz field theory (isotropic Heisenberg ferromagnet)

$$H_{\mathrm{LL}} = J \int \mathrm{d}x\, \mathbf{S}(x) \cdot \mathbf{S}_{xx}(x) = -J \int \mathrm{d}x\, \mathbf{S}_x(x)^2. \tag{239}$$

# E  Semi-classical limits of integrable quantum spin chains

## E.1  Time-dependent variational principle

We consider a single $N$-level quantum-mechanical degree of freedom in the Hilbert space $\mathbb{C}^N$, and a unitary time-evolution of a state $|\Psi\rangle = (\psi_0, \psi_1, \ldots, \psi_{N-1})^{\mathrm{T}}$ under Hamiltonian $\hat{H}$. By regarding $|\Psi(t)\rangle$ as a *variational* wavefunction, its time-evolution corresponds to extremizing a classical action $\mathcal{S} = \int \mathrm{d}t\, \mathcal{L}[\Psi]$ with Lagrangian $\mathcal{L}[\Psi(t)] = \langle \Psi(t) | \mathrm{i}\partial_t - \hat{H} | \Psi(t) \rangle$. This approach is called the *time-dependent variational principle*. In practice it is convenient to operate with an unnormalized wavefunction and treat the complex-conjugate field components $\bar{\psi}_j$ as independent variational variables. With this in mind, we consider a Lagrangian of the form

$$\mathcal{L}[\Psi, \bar{\Psi}] = \frac{\mathrm{i}}{2} \frac{\langle \Psi | \partial_t \Psi \rangle - \langle \partial_t \Psi | \Psi \rangle}{\langle \Psi | \Psi \rangle} - \frac{\langle \Psi | \hat{H} | \Psi \rangle}{\langle \Psi | \Psi \rangle}. \tag{240}$$

We next derive time evolution generated by $\hat{H}$ using generalized coherent states. For simplicity we restrict our considerations first to complex projective spaces $\mathbb{CP}^{N-1} = SU(N)/S(U(N-1) \times U(1))$ where, in close analogy to the well-known spin-coherent states of $\mathbb{CP}^1 \cong S^2$, we can define the $\mathbb{CP}^{N-1}$ coherent states by the 'vacuum rotation'

$$|\Psi\rangle = g|0\rangle, \tag{241}$$

where $g \in G = SU(N)$ and $|0\rangle$ is the highest-weight state of an irreducible finite-dimensional representation of $\mathfrak{g} = \mathfrak{su}(N)$. We shall consider here only the fundamental representation.

Note that while a general group element $g$ is fully determined by $N^2 - 1$ real parameters (e.g. Euler angles), the number of independent parameters which parametrize $|\Psi\rangle$ is actually smaller. This comes from the fact that the vacuum state $|0\rangle$ stays intact under $SU(N-1)$ rotations in the orthogonal complement of $|0\rangle$. This means that the unit complex vector $|\Psi\rangle$ lies on the real sphere $S^{2N-1}$. However, vectors which only differ by an overall $U(1)$ phase represent the same physical state and must thus be identified, meaning that $|\Psi\rangle$ is indeed an element of a coset space $S^{2N-1}/U(1) \cong \mathbb{CP}^{N-1}$, a manifold or real dimension $2(N-1)$. A general un-normalized variational wavefunction $|\Psi\rangle \in \mathbb{CP}^{N-1}$ is therefore parametrized by $N-1$ complex variables $z_i$, namely local affine coordinates of complex projective spaces, $\mathbf{z} = (z_1, z_2, \ldots, z_n)^{\mathrm{T}}$. The Lagrangian takes the form

$$\mathcal{L}(\mathbf{z}, \bar{\mathbf{z}}) = \frac{\mathrm{i}}{2} \sum_{i=1}^{n} \left( \dot{z}_i \frac{\partial}{\partial z_i} - \dot{\bar{z}}_i \frac{\partial}{\partial \bar{z}_i} \right) \mathcal{K}(\mathbf{z}, \bar{\mathbf{z}}) - H(\mathbf{z}, \bar{\mathbf{z}}), \tag{242}$$

where $\dot{\mathbf{z}} = \mathrm{d}\mathbf{z}/\mathrm{d}t$ and $\mathcal{K}(\mathbf{z}, \bar{\mathbf{z}})$ is the Kähler potential of $\mathbb{CP}^{N-1}$ corresponding to the logarithm of the normalization amplitude

$$\mathcal{K}(\mathbf{z}, \bar{\mathbf{z}}) = \log\langle\Psi(\mathbf{z})|\Psi(\mathbf{z})\rangle, \tag{243}$$

representing a hermitian metric tensor $\eta$ known as the Fubini–Study metric which can be produced via differentiation

$$\eta_{ij}(\mathbf{z}, \bar{\mathbf{z}}) = \frac{\partial^2 \mathcal{K}(\mathbf{z}, \bar{\mathbf{z}})}{\partial z_i \partial \bar{z}_j}. \tag{244}$$

Taking the variation

$$\delta S = \int \mathrm{d}t \left[ \sum_{i,j=1}^{n} \mathrm{i}(\dot{z}_i \eta_{ij} \delta\bar{z}_j - \dot{\bar{z}}_i \bar{\eta}_{ij} \delta z_j) - \delta H(\mathbf{z}, \bar{\mathbf{z}}) \right], \tag{245}$$

we deduce the following Euler–Lagrange equations

$$\sum_j \mathrm{i}\, \eta_{ji} \dot{z}_j = \frac{\partial H(\mathbf{z}, \bar{\mathbf{z}})}{\partial \bar{z}_i}, \qquad \sum_j \mathrm{i}\, \bar{\eta}_{ji} \dot{\bar{z}}_j = -\frac{\partial H(\mathbf{z}, \bar{\mathbf{z}})}{\partial z_j}. \tag{246}$$

Since the metric tensor is non-degenerate, $\mathrm{Det}(\eta) \neq 0$, Eqs. (246) can be readily inverted

$$\dot{z}_i = -\mathrm{i} \sum_j \left( \eta^{-1} \right)_{ji} \frac{\partial H}{\partial \bar{z}_j}. \tag{247}$$

The inverse of the Fubini–Study metric reads explicitly

$$\left( \eta^{-1} \right)_{ij} = (1 + \mathbf{z} \cdot \mathbf{z}) \left( \delta_{ij}(1 + z_j \bar{z}_i) + (1 - \delta_{ij}) z_j \bar{z}_i \right). \tag{248}$$

The above construction can be straightforwardly lifted to complex Grassmannian manifolds $\mathrm{Gr}_{\mathbb{C}}(k, N)$. In this case the variational wavefunction $\Psi$ becomes a complex matrix of dimension $N \times k$, whereas the Riemann metric tensor

$$\eta^{(k,N)} = \sum_{i,j=1}^{N-k} \sum_{a,b=1}^{k} \eta^{(k,N)}_{ia,jb} \mathrm{d}z_{i,a} \mathrm{d}\bar{z}_{j,b}, \tag{249}$$

can be again computed with aid of the Kähler potential $\mathcal{K}(Z, \bar{Z}) = \log\langle Z|Z\rangle$, where $\langle Z_1|Z_2\rangle = \mathrm{Det}(\mathbb{1} + Z_1^{\dagger} Z_2)$, reading

$$\eta^{(k,N)}_{ia,jb} = \left[ (\mathbb{1}_{N-k} + ZZ^{\dagger})^{-\mathrm{T}} \right]_{ij} \left[ (\mathbb{1}_k + Z^{\dagger}Z) \right]_{ab}. \tag{250}$$

## E.2 Coherent-state path integral

In this section we derive the effective classical action which governs the semi-classical eigenstates in integrable quantum chains invariant under global $SU(N)$ symmetry, described by Hamiltonians of the form

$$\hat{H} = \mathrm{J} \sum_{\ell=1}^{L} \left( \mathbb{1} - \Pi_{\ell,\ell+1} \right). \tag{251}$$

Here $\mathrm{J} > 0$ is the ferromagnetic exchange coupling and $\Pi |\alpha\rangle \otimes |\beta\rangle = |\beta\rangle \otimes |\alpha\rangle$ is the permutation matrix acting in $\mathbb{C}^N \otimes \mathbb{C}^N$, i.e. in the tensor product of two fundamental $\mathfrak{su}(N)$ representations. Here and subsequently we shall keep dependence on $N$ implicit throughout the derivation. This class of integrable models, introduced in [87,88], can be diagonalized by means of the Bethe Ansatz [13].

In the basis of traceless hermitian generators $X^a \in \mathfrak{g}$ (cf. Eqs. (33) and (34)), the permutation matrix assumes an expansion

$$\Pi = \frac{1}{N} + \sum_{a,b=1}^{N} \kappa_{ab} X^a \otimes X^b, \tag{252}$$

where $\kappa_{ab} \equiv 2\delta_{ab}$.

Computing semi-classical limit of Eq. (251) amounts to completely neglect quantum correlations in the variational wavefunction. This is achieved by projecting the Hamiltonian onto the subspace of many-body (product) coherent states

$$|\Psi(t)\rangle = \bigotimes_{\ell=1}^{L} |\Psi_{\ell}(t)\rangle, \tag{253}$$

where $|\Psi_{\ell}(t)\rangle$ is a $\mathbb{CP}^{N-1}$ coherent state inserted at position $\ell$. The variational states $|\Psi\rangle$ thus belong to the classical phase space $\mathcal{M}_L$ which parametrizes the low-energy sector of the quantum spin chain Hilbert space.

We now proceed with the path-integral computation. Fixing the 'initial' and 'final' states $|\Psi_i\rangle$ and $|\Psi_f\rangle$, respectively, the task at hand is to compute the quantum-mechanical transition amplitude

$$\mathcal{T}_t(\Psi_f, \Psi_i) = \langle \Psi_f| \exp(-\mathrm{i}\, t\, \hat{H}) |\Psi_i\rangle = \int \mathcal{D}[\Psi(t)] \exp\left( \mathrm{i}\, \mathcal{S}[\Psi(t)] \right), \tag{254}$$

and determine the classical action $\mathcal{S}[\Psi(t)]$. Slicing the time interval $[0, t]$ into N tiny intervals of size $\Delta t$ – with intermediate time coordinates $t_i = (t/\mathrm{N})i$ – and inserting the resolution of the identity,

$$\mathbb{1} = \int_{\mathcal{M}_L} \mathrm{d}\Omega^{(i)} |\Psi^{(i)}\rangle \langle \Psi^{(i)}|, \qquad |\Psi^{(i)}\rangle \equiv |\Psi(t_i)\rangle, \tag{255}$$

we find at the leading order $\mathcal{O}(\Delta t)$

$$\mathcal{T}_t(\Psi_\mathrm{f}, \Psi_\mathrm{i}) = \lim_{\Delta t \to 0} \prod_{i=0}^{N-1} \int_{\mathcal{M}_L} \mathrm{d}\Omega^{(i)} \, \langle \Psi^{(i+1)} | \, \mathbb{1} - \mathrm{i}\Delta t \, \hat{H} \, | \Psi^{(i)} \rangle . \tag{256}$$

The boundary conditions can be set to $|\Psi^{(N)}\rangle \equiv |\Psi_\mathrm{f}\rangle$ and $|\Psi^{(0)}\rangle \equiv |\Psi_\mathrm{i}\rangle$. Notice that the action involves two types of terms,

$$\mathcal{S} = \mathcal{S}_\mathrm{WZ} - \mathcal{S}_\mathrm{kin}. \tag{257}$$

The first term is a geometric contribution which stems from non-orthogonality of coherent states located at two adjacent time slices,

$$\langle \Psi^{(i+1)} | \Psi^{(i)} \rangle = \langle \Psi(t + \Delta t) | \Psi(t) \rangle = \exp\Big( - \Delta t \langle \Psi(t) | \partial_t \Psi(t) \rangle \Big) + \mathcal{O}(\Delta t^2), \tag{258}$$

which, in the $\Delta t \to 0$ limit, produces the Wess–Zumino term

$$\mathcal{S}_\mathrm{WZ} = \int \mathrm{d}t \, \mathcal{L}_\mathrm{WZ}, \qquad \mathcal{L}_\mathrm{WZ} = \mathrm{i}\langle \Psi | \partial_t \Psi \rangle. \tag{259}$$

The kinetic part $\mathcal{S}_\mathrm{kin} = \int \mathrm{d}t \int \mathrm{d}x \, \mathcal{L}_\mathrm{kin}(x)$ in Eq. (257) comes from spin interaction governed by the quantum Hamiltonian $\hat{H}$. To extract the corresponding Lagrangian density, let us initially specialize to $\mathcal{M}_1 \cong \mathbb{CP}^{N-1}$, where each coherent state can be represented by a rank-1 projector $P = |\Psi\rangle\langle\Psi|$. For computational purposes we instead define a hermitian traceless matrix

$$Y = N g |0\rangle \langle 0| g^\dagger - \mathbb{1}_N = N P - \mathbb{1}_N, \tag{260}$$

which is subjected to the non-linear constraint

$$Y^2 = (N-2)Y + (N-1)\mathbb{1}_N. \tag{261}$$

The expectation value of $\hat{H}$ in a many-body semi-classical state (253) is calculated as

$$\langle \hat{H} \rangle \equiv \mathrm{Tr}\left( \hat{H} \bigotimes_{\ell=1}^{L} Y_\ell \right) = JL\left( \frac{N-1}{N} \right)\mathbb{1} - 2J\sum_{\ell=1}^{L}\sum_a^N \langle X^a \rangle_\ell \langle X^a \rangle_{\ell+1}, \tag{262}$$

where coherent-state averages of the $\mathfrak{su}(N)$ generators read

$$\langle X^a \rangle_\ell \equiv \langle \Psi_\ell | X_\ell^a | \Psi_\ell \rangle = \mathrm{Tr}(X^a P_\ell) = \frac{1}{N}\mathrm{Tr}(X^a Y_\ell). \tag{263}$$

In order to take the continuum limit, we first recast Eq. (262) in the difference form. Exploiting the completeness relation $\sum_{a,b} \kappa_{ab} \mathrm{Tr}(X^a A)\mathrm{Tr}(X^b A) = \mathrm{Tr}(A^2)$, valid for an arbitrary traceless $N$-dimensional matrix $A$, we have the sum rule

$$2\sum_a [\mathrm{Tr}(X^a Y)]^2 = \mathrm{Tr}\, Y^2 = N(N-1). \tag{264}$$

This allows us to write

$$\langle \hat{H} \rangle = J\sum_{\ell=1}^{L}\sum_a \Big( \langle X^a \rangle_\ell - \langle X^a \rangle_{\ell+1} \Big)^2, \tag{265}$$

where we have used that $2\sum_a \langle X^a \rangle_\ell \langle X^a \rangle_{\ell+1} = 2\sum_a \langle X^a \rangle_\ell^2 - \sum_a (\langle X^a \rangle_\ell - \langle X^a \rangle_{\ell+1})^2$. With the aid of the long-wavelength expansion (assuming the lattice spacing $\Delta = 1/L$)

$$Y_\ell \to Y(x), \qquad Y_{\ell+1} \to Y(x) + \Delta \partial_x Y(x) + \frac{\Delta^2}{2}\partial_x^2 Y(x) + \dots, \tag{266}$$

we can finally pass to the field-theory regime. At the leading order $\mathcal{O}(\Delta^2)$ we deduce

$$\left[\text{Tr}\left(X^a(Y_\ell - Y_{\ell-1})\right)\right]^2 = \Delta^2\left[\text{Tr}(X^a Y_x) + \mathcal{O}(\Delta Y_{xx})\right]^2 \to \Delta^2\left[\text{Tr}(X^a Y_x)\right]^2, \tag{267}$$

where we have used the short-hand notation $Y_x \equiv \partial_x Y(x)$ and similarly for higher partial derivatives. Finally, replacing the sum $\sum_\ell$ with an integral $\int dx$ and introducing the renormalized coupling $J_{\text{cl}} = J\Delta^2/2$, we arrive at a compact final result

$$\langle \hat{H} \rangle = \frac{J_{\text{cl}}}{N^2} \int dx\, \text{Tr}\left(Y_x^2\right) = J_{\text{cl}} \int dx\, \text{Tr}\left(P_x^2\right). \tag{268}$$

The kinematic contribution to the Lagrangian density is therefore $\mathcal{L}_{\text{kin}}(x) = J_{\text{cl}}\text{Tr}((\partial_x P(x))^2)$.

To conclude with a simple example, in the case $\mathbb{CP}^1 \cong S^2$ we have $Y(x) \equiv M(x) = \boldsymbol{\sigma} \cdot \mathbf{S}(x)$, and using $\frac{1}{2}\text{Tr}(M_x^2) = \mathbf{S}_x \cdot \mathbf{S}_x$ we retrieve the Hamiltonian of the isotropic Landau–Lifshitz magnet,

$$\langle \hat{H} \rangle_{\mathbb{CP}^1} \equiv H_{\text{LL}} = \frac{J_{\text{cl}}}{2} \int dx\, \mathbf{S}_x(x) \cdot \mathbf{S}_x(x) = -\frac{J_{\text{cl}}}{2} \int dx\, \mathbf{S}(x) \cdot \mathbf{S}_{xx}(x), \tag{269}$$

with the equation of motion

$$\mathbf{S}_t = -\mathbf{S} \times \frac{\delta H_{\text{LL}}}{\delta \mathbf{S}} = J_{\text{cl}}\mathbf{S} \times \mathbf{S}_{xx}. \tag{270}$$

An explicit construction involving $SU(N)$ coherent states for $\text{Gr}_{\mathbb{C}}(k, N)$ manifolds is slightly more involved and for brevity we shall omit it. Note that the expectation values of the Lie algebra generators still correspond to the momentum maps, that is traces with respect to rank-$k$ projectors $P_\ell$. Up to a shift and rescaling, the latter is equivalent to the traceless hermitian matrix variable $Y = N P - k \mathbb{1}_N$, obeying $Y^2 = (N - 2k)Y + (N k - k^2)\mathbb{1}_N$. By repeating the above derivation, one once again arrives at Eq. (268), apart from a constant shift.

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
