# Peer review of "Integrable Matrix Models in Discrete Space-Time"

_SciPost Physics, doi:SciPost Phys. 9, 038 (2020)_

## Round 1 · Referee Report · Anonymous (Referee 1) · 2020-5-8

Strengths

A simple class of integrable dynamical systems with reasonably convincing numerical evidence of KPZ universality.

Weaknesses

1) Notations are not always consistent, see some examples below, which makes some parts of the paper hard to read without having to guess what the authors meant.

2) It is not clear which parts of the detailed analysis of the integrable structure of the model in section 2 is used for the numerical study of the large scale KPZ behavior in section 3.

Report

The authors introduce a new class of integrable dynamical systems describing the discrete time evolution of matrix valued fields located on a one-dimensional lattice with periodic boundary conditions. The invariant measure of the model and the integrability of the dynamics are studied in details.

The two-point correlation function of the models is then studied numerically for large values of time and number of lattice sites. Starting with equilibrium initial condition, the properly rescaled result is, after fitting amplitude and distance, in reasonable agreement with the stationary two-point function of the KPZ fixed point on the infinite line. Since the central part of the KPZ stationary two-point function is very close to a Gaussian, however, different choices of fitting parameters would presumably almost make it possible to fit the data obtained with a Gaussian too.

Given the simplicity of the models introduced and the distinct possibility that further studies manage to establish analytically that these models belong to KPZ universality, I think that this paper is perfectly suitable for publication in scipost.

Requested changes

1) It is not immediately clear that $L^{(+)}(\lambda, M)$ in (2.2) and $L_{n,m}^{(+)}(\lambda)$ in (2.1) are the same object. It would be useful to write down explicitly the relation between $L^{(+)}(\lambda, M_{l}^{t})$ and $L_{n,m}^{(+)}(\lambda)$.

2) The space $\mathcal{M}_{1}$ before (2.7) is not defined.

3) What is $\mathcal{Z}^{(k,N)}$ in (2.68) ? Is it the same as $\mathcal{Z}$ in (2.64) ?

4) The use of the Duistermaat–Heckman formula in section 2.3.3 should probably be detailed a bit.

5) It would be useful to specify precisely which parts of section 2 are used in section 3. Is the integrability of the model needed at all for the numerics ?

6) fig. 5: (d) $(2,N)=(1,4)$ ?

  • validity: high
  • significance: high
  • originality: high
  • clarity: ok
  • formatting: perfect
  • grammar: perfect

Author:  Enej Ilievski  on 2020-07-23  [id 903]

(in reply to Report 1 on 2020-05-08)
Category:
answer to question

We thank the referee for their assessment and for recommending the publication. Our response follows below:

"Since the central part of the KPZ stationary two-point function is very close to a Gaussian, however, different choices of fitting parameters would presumably almost make it possible to fit the data obtained with a Gaussian too."

While the central piece of the KPZ scaling function is difficult to distinguish from a Gaussian, the difference becomes apparent in the tails. This is the reason why sufficiently large simulation times (and system sizes) are required in order to reliably distinguish between the two. The main purpose of Section 3 is precisely to rule out the Gaussian profiles and to corroborate the KPZ scaling. In this regard, the constructed discrete-time dynamics helps appreciably.

The referee speculates whether it could be "almost possible" to fit a Gaussian to our data. Notice that In Fig. 7 and Fig. 8 we display "the best" (in the nonlinear least-squares sense) Gaussian fit onto the collapsed (nearly-converged) stationary profiles (marked by red dashed curves). One can observe tails that are widely separated from the data over a wide range of values on the y-axis. This data rules the Gaussian scaling out in a convincing fashion. We would like to argue that, to the best of our knowledge, our data constitutes the most convincing evidence for KPZ scaling in integrable models until this date.

Our replies to the specific queries:

1) We begin the discussion by shortly introducing the general framework of the auxiliary linear transport problem on a square lattice. For this purpose we write abstractly L^{(\pm)}_{n,m}, referring to the elementary propagators of the auxiliary light-cone lattice; lower indices designate the base point (n,m), while upper indices indicate the light-cone direction. Afterwards we introduce physical degrees of freedom, making the original notation with auxiliary lattice labels deprecated. We accordingly adapted the notation by suppressing unnecessary information. Uniformizing the notation here is in our opinion not an ideal solution as it would compromise clarity. We have added a brief clarifications (below Eq. 2.2) regarding our choice of notations in case of confusion.

2) Notice that $\mathcal{M}_{1}$ has been defined in the text as a "manifold of involutory matrices" (with no further requirements). For the specification of the symplectic map, this is indeed perfectly sufficient. It is only afterwards than we specialize our discussion to complex Grassmannian (or Lagrangian) manifolds, and these are discussed in more detail. While reversing the order of these two sections has been considered, it did appears to be an optimal way of presenting the concepts.

3) Yes, the two coincide. There we have suppressed the superscript as there was no room for ambiguity (see e.g. the integration measure in Eq. 2.64). We have now reinstated the superscripts to make the dependence on $k$ and $N$ explicitly visible.

4) Our perspective on this matter is perhaps a bit different. We would much rather prefer to stick with our initial "minimal" exposition at this stage, the main reason being any further clarification would necessitate a number of additional formal concept (all rather technical in nature) that could damage the presentation flow. Note that this particular computation is not an original result of ours, and also has no direct impact on what follows; it merely offers an elegant way for evaluating the phase-space integral without much hassle. The interested reader should be able to find more information in the included references, where the theory behind the localization theorem is discussed in greater detail.

5) Correct, integrability is not at all needed to carry out numerical simulations. In our numerics, integrability shows up in a subtle and indirect fashion through the emergence of the KPZ scaling. Notice that in the "Outline" paragraph, preceding Section 2, we indeed address precisely this point: Section 3 can be read as a "standalone", using only the dynamical map introduced at the start of Section 2.

6) We thank the referee for catching this misprint, which we have now fixed.

---

## Round 1 · Referee Report · Anonymous (Referee 2) · 2020-6-15

Strengths

  1. New results on the role of non-abelian symmetries for superdiffusive transport in 1D.
  2. Numerical results are convincing.
  3. Construction of a novel class of integrable matrix models.

Weaknesses

None.

Report

The authors introduce a (to the best of my knowledge) novel class of
integrable classical matrix models in discrete space time. These can
be viewed as Trotterizations of non-relativistic coset sigma
models. The models are integrable by virtue of a discrete
zero-curvature condition that guarantees the existence of infinitely
many conservation laws. The authors then carry out numerical studies
of transport properties in equilibrium by computing non-equal time
two-point functions of Noether charges. The numerics establishes in a
very convincing fashion that the dynamical exponent is z=3/2, in
agreement with KPZ superdiffusive behavior. They also find scaling
collapse of scaled "dynamical structure factors".

I think this is an excellent piece of work that makes an important
contribution to the current debate on KPZ universality in equilibrium
transport properties in one dimensional quantum and classical
many-body systems. In particular it provides strong support for the
conjecture formulated by the authors, namely that all discrete space-time
models built as Floquet circuits from two-body symplectic Yang-Baxter maps
with dynamical variables living on compact non-abelian symmetric
spaces exhibit superdiffusion of the KPZ type in equilibrium states
with unbroken symmetry. The work paves the way a work in a number of
interesting directions, e.g. the role of anisotropy. The manuscript is
very well written and I recommend publication in its current form.
  • validity: top
  • significance: top
  • originality: top
  • clarity: top
  • formatting: excellent
  • grammar: excellent

Author:  Enej Ilievski  on 2020-07-23  [id 902]

(in reply to Report 2 on 2020-06-15)
Category:
remark

We thank the referee for their assessment and encouraging words, and also for recommending the publication.

There indeed is plenty of room for generalizations and several interesting directions to pursue in the future, some of which we briefly outlined in the closing section.

---

## Round 2 · Referee Report · Anonymous (Referee 1) · 2020-8-13

Report

The authors took into account the remarks made by the referees. I recommend the paper for publication in SciPost.

---

## Round 2 · Author Response

Revised version.

---

## Round 2 · List of Changes

• We have addressed the points by one of the referees (short notational clarifications added when appropriate, fixed misprints).
  • We have resolved and a number misprints throughout the text.

---

## Editorial Decision

published